# Modelling snowpack on ice surfaces with the ORCHIDEE land surface model: Application to the Greenland ice sheet

Sylvie Charbit[1], Christophe Dumas[1], Fabienne Maignan[1], Catherine Ottlé[1], Nina Raoult[2], Xavier Fettweis[3], and Philippe Conesa[1]

[1]Laboratoire des Sciences du Climat et de l'Environnement, LSCE/IPSL, UMR 8212 CEA-CNRS-UVSQ, Université Paris-Saclay, 91191, Gif-sur-Yvette, France.
[2]Department of Mathematics and Statistics, Faculty of Environment, Science and Economy, University of Exeter, Laver Building, North Park Road, Exeter, EX4 4QE, United Kingdom.
[3]Laboratory of Climatology, Department of Geography, SPHERES, University of Liège, Liège, Belgium.

*Correspondence to*: Sylvie Charbit (sylvie.charbit@lsce.ipsl.fr)

**Abstract.** Current climate warming is accelerating mass loss from glaciers and ice sheets. In Greenland, the rates of mass changes are now dominated by changes in surface mass balance (SMB) due to increased surface melting. To improve the future sea-level rise projections, it is therefore critical to have an accurate estimate of the SMB, which depends on the representation of the processes occurring within the snowpack. The snow scheme (ES) implemented in the land surface model ORCHIDEE has not yet been adapted to ice-covered areas. Here, we present the preliminary developments we made to apply the ES model to glaciers and ice sheets. Our analysis mainly concerns the model's ability to represent ablation-related processes. At the regional scale, our results are compared to the MAR regional atmospheric model outputs and to MODIS albedo retrievals.

Using different albedo parameterizations, we performed offline ES simulations forced by the MAR model over the 2000-2019 period. Our results reveal a strong sensitivity of the modeled SMB components to the albedo parameterization. Results inferred with albedo parameters obtained with a manual tuning approach present a very good agreement with the MAR outputs. Conversely, with the albedo parameterization used in the standard ORCHIDEE version, runoff and sublimation were underestimated. We also tested parameters found from a previous data assimilation experiment calibrating the ablation processes using MODIS snow albedo. While these parameters greatly improve the modelled albedo over the entire ice sheet, they degrade the other model outputs compared to those obtained with the manually-tuned approach. This is likely due to the model overfitting to the calibration albedo dataset without any constraint applied to the other processes controlling the state of the snowpack. This underlines the need for performing a "multi-objective" optimisation using auxiliary observations related to snowpack internal processes. Although there is still room for further improvements, the developments reported in the present study constitute an important advance in assessing the Greenland SMB with possible extension to mountain glaciers or the Antarctic ice sheet.

## 1. Introduction

Satellite observations reveal that the Greenland ice sheet (GrIS) has been losing mass for at least three decades. Between 1992 and 2018, the net ice mass loss was estimated at 3800 ± 339 Gt, corresponding to a rise in global mean sea level of 10.6 ± 0.9 mm (The IMBIE team, 2020). Mass loss is driven by dynamic solid ice discharges (Enderlin et al., 2014) and by enhanced surface meltwater and runoff (Ryan et al., 2019). Over the 2000-2008 period, the GrIS mass loss was equally partitioned between surface and dynamic processes (van den Broeke et al.,

2009). However, recent studies based on regional climate models and remote sensing observations (van den Broeke, 2016; Ryan et al., 2019; The IMBIE Team, 2020, Fox-Kemper et al., 2021) show that rates of mass change are now dominated by changes in surface mass balance (SMB), defined as the difference between mass gains (solid and liquid precipitation) and surface ablation processes (runoff, sublimation and snow erosion).

Besides directly impacting the global mean sea level, the GrIS is also an integral part of the Earth System (Fyke et al., 2018). As such, it is highly sensitive to climate change and in turn, has a strong influence on global climate, notably by releasing fresh water into the ocean, which leads to changes in the Atlantic meridional overturning circulation (Bakker et al., 2016; Martin et al., 2022). Surface melting may also induce changes in the local climate through the temperature-elevation feedback (Edwards et al., 2014; Sellevod et al., 2019) and the albedo effect (Box et al., 2012; Helsen et al., 2017; Riihelä et al., 2019). Finally, changes in topography produce modifications of the local and large-scale atmospheric circulations (Ridley et al., 2005; Hahn et al., 2020).

To capture this feedback and to reduce the uncertainties in sea-level and climate projections, a key objective of the climate-ice sheet modelling community is to incorporate ice-sheet models in Earth System Models (ESMs) (Vizcaino, 2014). Such coupled climate-ice sheet models have mainly been developed with low resolution climate models designed for long-term integrations (Kageyama et al., 2004; Charbit et al., 2005; Vizcaino et al., 2010; Roche et al., 2014). So far, only a few groups have met this goal with CMIP-like models (Vizcaino et al., 2013; Muntjewerf et al., 2020; Smith et al., 2021). A key challenge in developing such models relates to the realistic computation of SMB used as a forcing field of the ice-sheet models.

SMB is highly dependent on the radiative properties of snow and on the physical processes occurring within the snowpack (Helsen et al., 2017). At the surface, snow cover evolves as a function of the surface energy balance and mass exchanges with the atmosphere. In cold regions, snow melt is largely driven by shortwave radiation: Because of the high albedo value of fresh snow (0.80 – 0.90), a large fraction of shortwave radiation is reflected to the atmosphere, limiting the energy available at the surface for melting. Therefore, snow evolution is strongly dependent on the albedo. The value of snow albedo decreases when snow is ageing (i.e. in the absence of a new snowfall event) and with the snow metamorphism and liquid water content at the ice sheet's surface coming either from rainfall or from snow/ice melting. Surface water may also percolate and refreeze inside the snowpack, thereby delaying the runoff. The transformation of snow into ice depends on environmental conditions (e.g. winds, near-surface temperatures) and internal processes within the snowpack (e.g. heat conduction and vertical temperature gradient, compaction), which directly influence the grain microstructure and the snow density. All these processes affect the SMB of the ice sheet.

There are several ways to compute the SMB. Empirical approaches such as the positive degree-day method (Reeh, 1991) have long been used to compute snow and ice melting from downscaled near-surface temperatures. This kind of approach requires little computational resources and has often been applied for past and future long-term integrations (Charbit et al., 2008; 2013; Bonelli et al., 2009; Vizcaino et al., 2010). However, such methods have been calibrated against the present state of the GrIS, raising the question as to whether they can be applied in a different climatic context from the present-day one knowing that ablation is projected to increase (van de Wal, 1996; Bougamont et al., 2007). Moreover, they are not physically-based and cannot reproduce the diversity of snow processes that directly influence the SMB. Snow models implemented in general circulation models have long been based on simplified physics. They are mainly designed to resolve the seasonal and diurnal variations of heat fluxes, but with no representation of internal processes (Armstrong and Brun, 2008). By contrast, regional

climate models developed for polar regions generally incorporate multiple-layer energy balance snow models with
a fine vertical resolution (e.g. Brun et al., 1992; Lefebre et al., 2003; Vionnet et al., 2012; Noël et al., 2018) and
with detailed snow physics to simulate a variety of snowpack processes. However, due to their high computational
cost, they are not often used in ESMs, despite a few rare exceptions such as the work of Punge et al. (2012) based
on the implementation of a detailed snow model (Brun et al., 1992) in the atmospheric model LMDZ4 (Hourdin
et al., 2006), or the Community Land Model (CLM) which includes the snow radiative transfer scheme SNICAR
(Flanner and Zender, 2006) and a snow model simulating a variety of key snow processes such as the
metamorphism (Lawrence et al., 2019, He et al., 2024).
An alternative approach consists in implementing snow models of intermediate complexity in the land surface
components of ESMs (Boone and Etchevers, 2001; Dutra et al., 2010; Wang et al., 2013; Cullather et al., 2014;
Decharme et al., 2016; Born et al., 2019). These models have a limited number of layers and are based on simplified
representations of the main processes affecting the SMB changes, but usually do not have any explicit
representation of snow metamorphism. However, they offer a good compromise between models of high
complexity and simplified approaches or bulk-layer models for coupling with atmospheric models.
The snow module Explicit Snow (referred hereafter to as ES) implemented in the land surface model ORCHIDEE
(Organising Carbon and Hydrology In Dynamic Ecosystems; Krinner et al., 2005; Chéruy et al., 2020) of the
IPSL-CM ESM (Boucher et al., 2020) belongs to this third class of snow models. It has been successfully evaluated
against observations in Col de Porte (French Alps) and in various sites of Northern Eurasia (Wang et al., 2013).
However, it has not yet been adapted to ice-covered areas. As a result, glaciers are considered as bare soils in the
current ORCHIDEE version, and over ice sheets, snow is handled with the atmospheric component of IPSL-CM
in a very simplistic way. Recently, we made new developments to apply the ES model to glaciers and ice sheets,
with a special focus on the GrIS. These developments meet two objectives. The first one is to treat snow-related
processes in IPSL-CM in a more consistent way for all surface types. The second one is to compute the SMB,
taking the main processes occurring within the snowpack into account. These developments also constitute a
preliminary step for the subsequent use of the computed SMB as an interface between IPSL-CM and ice-sheet
models. In the following, we will refer to ORCHIDEE-ICE to deal with the version of ORCHIDEE that includes
these new developments, and to ORCHIDEE to deal with the former version of the model.
In this study, we evaluate the computation of SMB (and its components) in the ES model. As SMB is strongly
dependent on the albedo, we also examine its sensitivity to various albedo parameterizations. To achieve this, we
performed offline ORCHIDEE-ICE simulations and compared our results against model outputs from the polar-
oriented regional atmospheric model MARv3.11.4 (Modèle Atmosphérique Régional, Fettweis et al., 2017) and
the MODIS (MODerate resolution Imaging Spectroradiometer, Hall et al., 1995; Hall and Riggs, 2016) surface
albedo retrievals. The paper is organized as follows. In Section 2, we provide an extensive description of the main
characteristics of the original ES model as well as changes that occurred since its early publication (Wang et al.,
2013). The new developments made for applying ES to the GrIS are also presented in this section. Section 3
describes the experimental setup and Section 4 provides a brief overview of the different datasets used for
evaluation. The results are presented in Sections 5 and 6 and discussed in Section 7.

## 2. Model description

### 2.1 Snow processes in the current ORCHIDEE-AR6 model

ORCHIDEE is the land surface component of the IPSL-CM Earth System Model (Boucher et al., 2020; Chéruy et al., 2020) mainly developed at the French Institute Pierre Simon Laplace (IPSL). It computes both the water and energy exchanges (SECHIBA module) between land surfaces and the atmosphere at a half-hourly time step and includes carbon-related processes (STOMATE module). Within a given grid cell, land cover is represented as fractions of bare soils and vegetated areas described in terms of plant functional types (PFTs). The snow-vegetation interactions are not explicitly represented and snow is evenly distributed among the various PFTs. Soil types are prescribed according to the USDA soil texture maps (Reynolds et al., 2000). The ORCHIDEE model can be run in off-line mode, driven by atmospheric fields, or coupled with an atmospheric model. In the former ORCHIDEE version used for CMIP5 (Taylor et al., 2012), the snow scheme over glaciated surfaces was based on the bulk approach proposed by Chalita and Le Treut (1994). It consisted of a composite soil-snow model accounting for the thermal and radiative properties of snow cover (i.e. albedo and its variations with snow ageing). Snow was described as having a constant density (330 kg m$^{-3}$) and melting occurred when temperature exceeded 0°C. Other processes such as water percolation and refreezing were ignored, although they directly impact the water budget. This means that all liquid water coming from melting snow was leaving the snowpack as runoff.

For the CMIP6 exercise (Eyring et al., 2016), the bulk approach has been replaced by the ES snow scheme, which was formerly adapted to the ORCHIDEE architecture (Wang et al., 2013) from a three-layer version of the ISBA-ES scheme (Interactions between Soil, Biosphere and Atmosphere-Explicit Snow scheme; Boone and Etchevers, 2001) developed at the French National Center for meteorological Research. The ES model is now used in the standard version of ORCHIDEE (version 2.0 onwards). However, it has not yet been considered for use over mountainous glaciers, which are treated as bare soils, nor over ice sheet areas, which are currently handled by the LMDZ atmospheric model (Chéruy et al., 2020) with a very elementary snow scheme (i.e. single-layer model, constant albedo and thermal conductivity). In this section, we provide an extensive description of the snow model, including the main differences with the original ISBA-ES version (Wang et al., 2013). The new developments accounting for snow processes over ice-covered areas in the ORCHIDEE model are described in section 2.2.

The ES model represents the snowpack as a one-dimensional physical system (vertical coordinate z). This means that all the lateral fluxes of mass and energy are ignored. The original version of this snowpack is discretized in three layers following the parameterization of Lynch-Stieglitz (1994), which sets the upper limits for the thickness of the first two layers at 5 and 50 cm respectively. This ensures the propagation of variations in the diurnal cycle of temperature and radiation, and enables vertical heat and density gradients, which are assumed to be larger near the surface, to be resolved correctly. Each layer is described in terms of snow density, snow age, layer thickness, heat content, snow temperature and liquid water content, with the first three variables being prognostic variables. Changes in snow mass are determined by the snowfall rate, snow melting, runoff at the base of the snowpack and sublimation at the surface. In the absence of coupling with a dynamic ice sheet model, snow mass at the surface of the ice sheet can be overestimated. Thus, to prevent excessive snow accumulation, we impose a maximum threshold of 3000 kg m$^{-2}$ beyond which snow is artificially removed. An overview of the organization of the different subroutines of the ES snowpack model is provided in Figure 1. The description of the processes is given in the following subsections and the list of model parameters is provided in Table A1 (Appendix A).

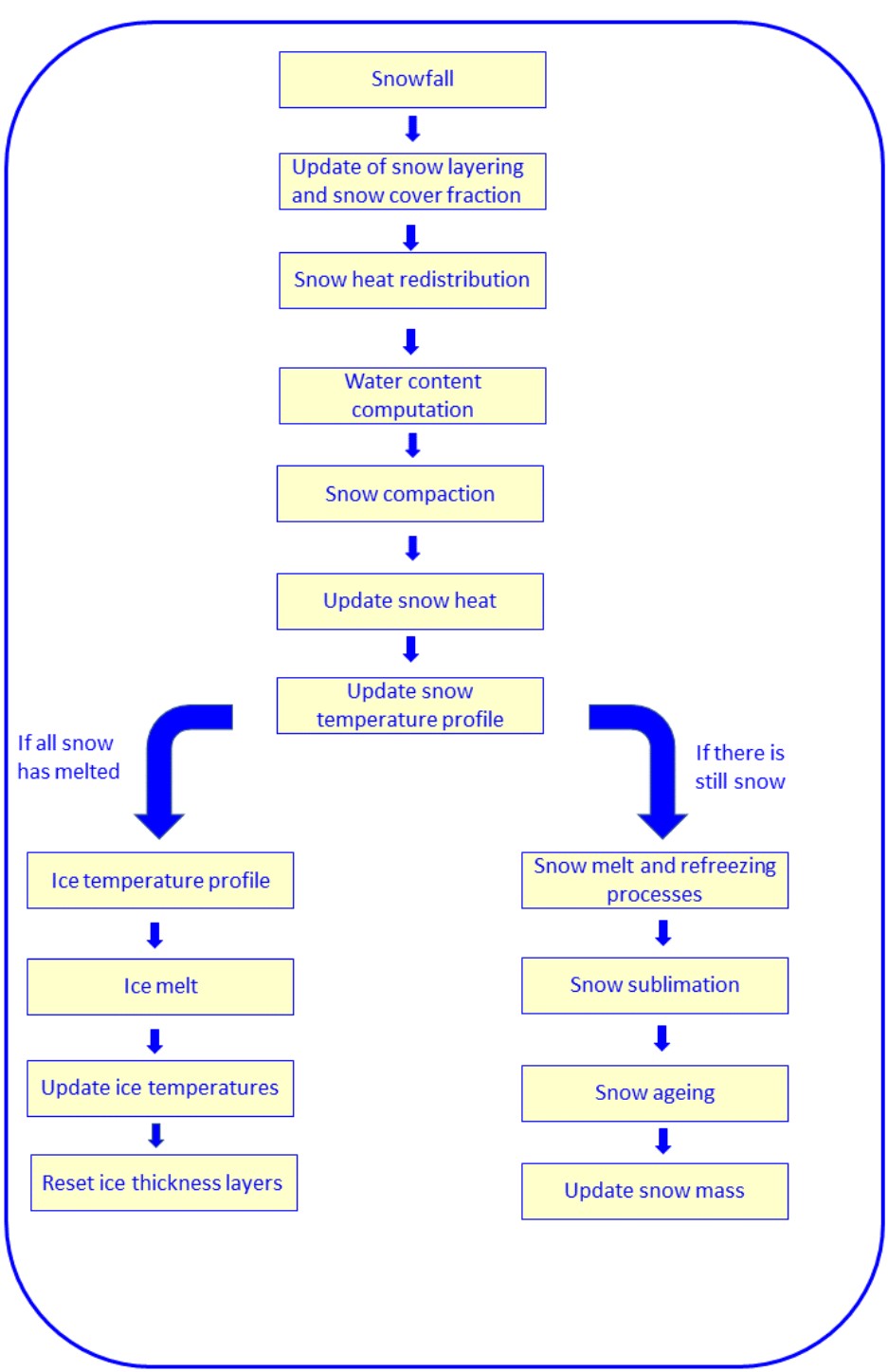


**Figure 1:** Flowchart of the new Explicit Snow scheme implemented in the ORCHIDEE-ICE model.




**2.1.1 Surface processes**
*Energy balance*
The evolution of the snowpack is primarily driven by the energy flux at the snow-atmosphere interface. A single
energy balance is computed for all surface types coexisting in one grid cell. The surface energy flux ($G_{surf}$)
available at the snow-atmosphere interface is computed from the energy balance equation:
$$G_{surf} = SW_{net} + LW_{net} - H_L - H_S + H_{rainfall} \tag{1}$$
$G_{surf}$ is computed positively when it warms the soil. $SW_{net}$ and $LW_{net}$ are the net shortwave and longwave
radiations respectively, $H_L$ is the latent heat flux, $H_S$ is the sensible heat flux and $H_{rainfall}$ is the energy released
by rainfall (see Eq. (14) in Boone and Etchevers, 2001). Equation (1) is used to compute the surface temperature
($T_{surf}$) of the grid cell at the next time step and provides the limit condition of the surface temperature at the snow-
atmosphere interface for the calculation of the snow temperature profile.
Above snow-covered surfaces, when $T_{surf}$ is above the freezing temperature $T_0$ (273.15 K), the energy excess is
first used to bring the snow temperature to $T_0$. A surface energy flux $G_{freezing}$ associated with the freezing
temperature $T_0$ can be computed using a similar formulation to Eq. (1). The difference between $G_{surf}$ and $G_{freezing}$
is converted in an additional temperature expressed as:
$$T_{snow}^{add} = T_{surf} - T_0 = \frac{G_{surf} - G_{freezing}}{C_{soil}} \, dt \tag{2}$$
$C_{soil}$ is the surface heat capacity of soil (J m$^{-2}$ K$^{-1}$) and is computed as the sum of heat capacities for snow-covered
and snow-free surfaces (for both non-glaciated and glaciated areas) weighted by their respective grid cell fractions.
For snow-covered surfaces, the specific heat capacity is defined as the product of snow density and the specific
heat of ice (2106 J K$^{-1}$ kg$^{-1}$). If $T_{snow}^{add}$ is greater than (or equal to) the freezing temperature, the energy excess is
used for melting snow, and $G_{surf}$ is further set to $G_{freezing}$ for energy conservation. If the new $G_{surf}$ value is
greater than the total heat content of the snowpack, snow is entirely melted and the excess energy is transferred to
the underlying soil. The energy released by snowfall is accounted for in the snowpack scheme to update the snow
heat content of the snowpack after a snowfall event.
*Turbulent heat fluxes*
The sensible ($H_S$) and latent heat ($H_L$) fluxes computed for each grid cell are given respectively by:
$$H_S = \rho_{air} q_{cdrag} U (T_{surf} - T_{air}) \tag{3}$$
$$H_L = L_s \rho_{air} q_{cdrag} U (Q_{sat} - Q_{air}) \tag{4}$$
where $\rho_{air}$ is the air density, $T_{surf}$ and $T_{atm}$ are the surface and the 2 m atmospheric temperatures, $Q_{air}$ and $Q_{sat}$
are the air specific humidity at 2 m and the saturated specific humidity at the surface, $L_s$ is the latent heat of
sublimation (2.8345 10$^6$ J kg$^{-1}$), $U$ is the wind speed at 10 m and $q_{cdrag}$ is the drag coefficient computed as a
function of the ice roughness length (*z0_ice* = 0.001 m), following the Monin-Obukhov turbulence theory (Monin
and Obhukov, 1954) and the parameterizations of the eddy fluxes proposed by Louis (1979).


*Snow sublimation*
The amount of sublimation is simply deduced from the latent heat flux:
$$S_{snow} = \frac{H_L}{L_s}$$ (5)
*Snow cover fraction*
The snow cover fraction ($F_{snow}$) is derived from the formulation of Niu and Yang (2007) which has been shown
to better represent the seasonal variation of the relationship between snow depth ($Z_{snow}$) and snow cover fraction
thanks to its dependence on snow density:
$$F_{snow} = \tanh\left(\frac{Z_{snow}}{2.5 z_{0g} \times \left(\frac{\langle \rho_{snow} \rangle}{\rho_{min}}\right)^m}\right)$$ (6)
where $\langle \rho_{snow} \rangle$ is the snow density averaged over the total thickness of the snowpack, $\rho_{min}$ is the minimum snow
density (set to 50 kg m$^{-3}$), that is the density of fresh snow, $z_{0g}$ is the ground roughness length (set to 0.01 m) and
m (set to 1.0 in the present study) is an adjustable parameter.
*Snow albedo*
Compared to the early version presented in Wang et al. (2013), the albedo scheme has been modified and snow
albedo is now computed following the formulation of Chalita and Le Treut (1994):
$$\alpha_{snow} = A_{aged} + B_{dec} exp\left(-\frac{\tau_{snow}}{\tau_{dec}}\right)$$ (7)
where $A_{aged}$ represents the albedo of a snow-covered surface after snow ageing (old snow) and $B_{dec}$ is defined so
that the sum of $A_{aged}$ and $B_{dec}$ represents the albedo of fresh snow (i.e. maximum snow albedo). $\tau_{dec}$ is the time
constant of the albedo decay and $\tau_{snow}$ is the snow age and is parameterized as follows:
$$\tau_{snow}(t + dt) = \left[\tau_{snow}(t) + \left(1 - \frac{\tau_{snow}}{\tau_{max}}\right) \times dt\right] \times exp\left(-\frac{P_{snow}}{\delta_c}\right) + f_{age}$$ (8)
where $\tau_{max}$ is the maximum snow age, $P_{snow}$ is the amount of snowfall during the time interval $dt$ and $\delta_c$ is the
critical value of solid precipitation necessary for reducing the snow age by a factor 1/e. As the ORCHIDEE time
step is fixed to 30 mn,, the snow age is almost zero in a few time steps. In addition, low surface air temperatures
found in polar regions slow down the metamorphism. This effect is accounted for with the function $f_{age}$ expressed
as:
$$f_{age} = \left[\frac{\left(\tau_{snow}(t) + \left(1 - \frac{\tau_{snow}}{\tau_{max}}\right) \times dt\right) \times exp\left(-\frac{P_{snow}}{\delta_c}\right) - \tau_{snow}(t)}{1 + g_{temp}(T_{surf})}\right]$$ (9)
$$g_{temp}(T_{surf}) = \left[\frac{max(T_0 - T_{surf}, \ 0)}{\omega_1}\right]^{\omega_2}$$ (10)
where $\omega_1$ and $\omega_2$ are tuning constants. The albedo is computed for the visible and near-infrared spectral bands.
However, to compute the upward shortwave radiation, an arithmetic mean between the visible and the near-
infrared albedo is considered.
A single energy balance is computed for all surface types but the albedo is weighted by the different fractions of
PFTs and glaciated areas and by the snow-covered and snow-free fractions. As a result, the surface albedo ($\alpha$) of
the grid cell is computed as the sum of snow-free albedo ($\alpha_{snow-free}$) and snow-covered albedo ($\alpha_{snow}$) weighted
by the fractional area of the grid cell covered by snow $F_{snow}$ (snow-covered fraction hereafter):
$\alpha = F_{snow} \times \alpha_{snow} + (1 - F_{snow}) \times \alpha_{snow-free}$                  (11)
with:
$\alpha_{snow} = f_{ice} \times \alpha_{snow}^{ice} + \sum_{PFT} f_{PFT,i} \times \alpha_{snow}^{PFT,i}$              (11a)
and:
$\alpha_{snow\_free} = f_{ice} \times \alpha_{snow\_free}^{ice} + \sum_{PFT} f_{PFT,i} \times \alpha_{snow\_free}^{PFT,i}$        (11b)
$f_{ice}$ and $f_{PFT,i}$ are the grid cell fractions of ice-covered areas and the $i^{th}$ PFT respectively; $\alpha_{snow}^{ice}$ (resp. $\alpha_{snow\_free}^{ice}$)
and $\alpha_{snow}^{PFT,i}$ (resp. $\alpha_{snow\_free}^{PFT,i}$) are the corresponding snow albedo (resp. snow-free albedo) values.
Over the GrIS, $\alpha_{snow-free}$ is given by the albedo of bare ice, prescribed to 0.6 and 0.2 for visible and near-infrared
wavelengths respectively. At the margins of the GrIS, some grid points may be only partially covered by snow or
ice, or even become totally snow-free during the melting season. It is therefore important to take these different
features into account to compute correctly the surface albedo of the GrIS.
**2.1.2 Internal processes**
When snow falls on a snow-free surface, a new snowpack is generated providing that the ground temperature is
below or equal to the freezing point. The snow mass and the heat content of the snowfall are initially distributed
evenly within the three layers. The snow density is the same for the three layers and is given by the density of the
snowfall computed as a function of wind speed and surface air temperature (Pahaut, 1976). When snowfall occurs
over an existing snowpack, fresh snow is added to the upper layer providing that the snowfall thickness is greater
than the critical threshold $\delta_c$ (see Eq. 8). The snow thickness, density and heat content are then modified in this
layer. However, as the number of snow layers is kept fixed, redistribution of mass and heat content within the
layers is required when snow depth changes, but the total snow mass and heat content are conserved.
*Heat conduction*
Solar absorption is not accounted for in the snow model. All incoming solar energy is therefore deposited at the
snow surface and distributed in deeper layers through heat conduction. The heat conduction from the surface to
the bottom of the snowpack is described by a vertical diffusion equation relating the temporal evolution of the
snow temperature in the snowpack at a depth $z$ and the divergence of the snow heat flux $F_C$ and is solved using an
implicit numerical scheme.
$\frac{\partial T_{snow}}{\partial t} = -\frac{1}{C_{snow}} \cdot \frac{\partial F_C}{\partial z}$                       (12)
$F_C = -\Lambda_s \frac{\partial T_{snow}}{\partial z}$                          (13)
with $C_{snow}$ (J m$^{-2}$ K$^{-1}$) $\Lambda_s$ and $T_{snow}$ being the snow heat capacity, the snow thermal conductivity (W m$^{-1}$ K$^{-1}$) and
the snow temperature respectively.
At the snow-atmosphere interface, the boundary condition is given by the energy balance equation ($F_c = G_{surf}$)
and is used in the ORCHIDEE model to compute the surface temperature.
Along with the thermal gradient, a water vapor diffusive flux takes place from the warmer to the colder parts of
the snowpack and sublimation or condensation may occur in the pore spaces depending on the water vapor
saturation pressure. This process is particularly significant in the Arctic because of strong temperature gradients
between soils and atmosphere and is in great part responsible for snow metamorphism. While it is explicitly
accounted for in detailed snow models, in Explicit Snow, the effect of water vapor diffusion and phase changes is
parameterized through the thermal conductivity (Sun et al., 1999). An effective thermal conductivity ($\Lambda_{eff}$) is thus
expressed as the sum of empirical formulations for snow thermal conductivity ($\Lambda_{cond}$) and thermal conductivity
from vapor transport ($\Lambda_{vap}$), with:
$$\Lambda_{cond}^{i} = a_\lambda + b_\lambda {\rho_{snow}^{i}}^{2} \tag{14}$$
$$\Lambda_{vap}^{i} = \left(a_{\lambda v} + \frac{b_{\lambda v}}{c_{\lambda v} + T_{snow}^{i}}\right)\frac{P_0}{P} \tag{15}$$
With $a_\lambda$= 0.02 W m$^{-1}$ K$^{-1}$, $b_\lambda$ = 2.5 10$^{-6}$ W m$^5$ K$^{-1}$ kg$^{-2}$ (Anderson, 1976), $a_{\lambda v}$ = -0.06023 Wm$^{-1}$K$^{-1}$, $b_{\lambda v}$ = -2.5425
W m$^{-1}$ and $c_{\lambda v}$ = -289.99 K (Yen, 1981). $P$ is the atmospheric pressure in hPa and $P_0$ = 1000 hPa. The superscripts
$i$ denote the $i^{th}$ layer.
***Heat content***
The heat content is computed using the following equation:
$$H_{snow}^{i} = D_{snow}^{i}\left[C_{snow}^{v,i}\left(T_{snow}^{i} - T_f\right) - L_s\rho_{snow}^{i}\right] + L_f\rho_{water}W_{liq}^{i} \tag{16}$$
where $L_f$ is the latent heat of fusion and $\rho_{water}$ is the water density. $H_{snow}^{i}$, $W_{liq}^{i}$, $D_{snow}^{i}$, $\rho_{snow}^{i}$ and $C_{snow}^{v,i}$ are
the heat and liquid contents, the depth, the density and the mean volumetric heat capacity (J K$^{-1}$ m$^{-3}$) of the $i^{th}$
layer.
After heat redistribution within the snowpack, snow temperature is diagnosed using Eq. (16), assuming no liquid
water in the snowpack. If snow temperature exceeds the freezing point, the liquid content in each layer is then
diagnosed from the snow temperature and heat content of the layer, and the temperature is then reset to the freezing
point.
***Compaction***
The total snow depth decreases as density increases. Changes in density occur as a result of the weight of the
overlying snow layers and under the influence of snow metamorphism. The local rate of density change in the $i^{th}$
layer is derived from Anderson (1976):
$$\frac{1}{\rho_{snow}^{i}}\frac{\partial \rho_{snow}^{i}}{\partial t} = \frac{\sigma_{snow}^{i}}{\eta_{snow}^{i}\left(T_{snow}^{i}, \rho_{snow}^{i}\right)} + \psi_{snow}^{i}\left(T_{snow}^{i}, \rho_{snow}^{i}\right) \tag{17}$$
The first term of the right-hand side represents the compaction due to snow load, with $\sigma_{snow}^{i}$ (Pa) being the pressure
of the overlying snow and $\eta_{snow}^{i}$ the snow viscosity.
$$\sigma_{snow}^{i} = g \ x \ M_{snow}^{i}$$
where $g$ is the gravitational constant (m s$^{-2}$) and $M_{snow}^{i}$ the cumulative snow mass (kg m$^{-2}$).
The viscosity (in Pa s) is expressed as a function of snow temperature and density (Mellor, 1964; Kojima, 1967):
$$\eta_{snow}^{i} = \eta_0 exp\left[a_\eta\left(T_f - T_{snow}^{i}\right) + b_\eta\rho_{snow}^{i}\right] \tag{18}$$
with $\eta_0 = 3.7 \times 10^7$ Pa s, $a_\eta = 8.1 \times 10^{-2}$ K$^{-1}$ and $b_\eta = 1.8 \times 10^{-2}$ m$^3$ kg$^{-1}$.
The second term in the right-hand side of Eq. (17) parameterizes the effect of metamorphism which is significant
for newly fallen snow.
$$\psi_{snow}^i = a_\psi exp\left[- b_\psi \cdot \left(T_f - T_{snow}^i\right) - c_\psi \cdot max\left(0, \rho_{snow}^i - \rho_\psi\right)\right] \tag{19}$$
The values of the parameters are the following: $a_\psi = 2.8 \times 10^{-6}$ s$^{-1}$, $b_\psi = 4.2 \times 10^{-2}$ K$^{-1}$, $c_\psi = 460$ m$^3$kg$^{-1}$, $\rho_\psi = 150$
kg m$^{-3}$.
In the model, density changes due to compaction are allowed as long as density remains below a threshold fixed
to 750 kg m$^{-3}$. This value was chosen because compaction becomes slower above densities between 550 and 800
kg m-3 due to the progressive disappearance of air spaces between the snow particles (Maeno and Ebinuma, 1983).
A critical value of 730 kg m-3 has even been advanced by Maeno (1978). Compaction does not affect the total
mass and the heat content of the snowpack but changes the layer thicknesses. The distribution of snow heat within
the layers must therefore be updated using Eq. (16).
*Vertical temperature profile*
The snow temperature profile resulting from heat redistribution is then computed by solving the heat diffusion
equation using an implicit numerical scheme similar to that used for heat diffusion in the soil. The vertical
temperature profile within the snowpack is expressed as:
For the 1$^{st}$ layer:
$$T_{snow}^1 = \left[\frac{\lambda_{snow} \cdot C_{gr\_snow} + \left(T_{surf} + T_{snow}^{add}\right)}{1 + \lambda_{snow}\left(1 - D_{gr\_snow}\right)}\right] \tag{20}$$
For the deeper layers (i > 1):
$$T_{snow}^{i+1} = C_{gr\_snow} + D_{gr\_snow} \cdot T_{snow}^i \tag{21}$$
where $\lambda_{snow}, C_{gr\_snow}, D_{gr\_snow}$ are coefficients resulting from the resolution of the numerical scheme and depend
on the snow heat capacity, thermal conductivity and characteristics of the vertical discretization. The numerical
scheme is similar to the one presented in Wang et al. (2016, see Appendix A therein) in which the temperature at
the interface between two layers is calculated as a linear interpolation according to the two nearest nodes (middle
of the layers). Diffusion therefore takes place downward and upward.
*Melting and refreezing processes*
If melt water is produced at the surface, it may remain in the liquid state in the uppermost layer or penetrate the
next layer where it can remain or refreeze as long as the maximum water holding capacity is not reached; otherwise,
it penetrates the lower layers.
The evolution of liquid water in each layer is controlled by the energy available to induce phase changes and by
the maximum water holding capacity. In the $i^{th}$ layer, the energy used for melting snow ($E_{snow}^i$) is expressed as:
$$E_{snow}^i = min\left( C_{snow}^{v,i} D_{snow}^i \times max\left(0, T_{snow}^i - T_f\right), max\left(0, D_{swe}^i - W_{liq}^i\right) \times L_f \rho_{water}\right) \tag{22}$$
where $D_{swe}^i$ is the snow water equivalent in the $i^{th}$ layer. The first term represents the available energy for phase
change in the $i^{th}$ layer and the second term corresponds to the energy required to melt entirely the snow mass that
has not been transformed into liquid water. The maximum water holding capacity is taken from Anderson (1976):
$$W_{max}^i = \left[ r_{min} + (r_{max} - r_{min}) \cdot max \left( 0, \frac{\rho_t - \rho_{snow}^i}{\rho_t} \right) \right] \cdot \frac{\rho_{snow}^i}{\rho_w} \cdot D_{snow}^i \tag{23}$$
with $r_{min} = 0.03$, $r_{max} = 0.10$ and $\rho_t = 200$ kg m$^{-3}$.
Runoff ($S_{melt}$) is computed as the sum of meltwater produced at the surface and the total liquid water that has
percolated down to the bottom layer and that exceeds $W_{max}^{bottom}$. It is thus simply given by:
$$M_{snow} = \frac{\sum_i E_{snow}^i}{L_f} \tag{24}$$
At each time step, changes in layer thickness, density and liquid water content in each layer are updated as well as
changes in snow temperature due to melting or refreezing. In case of complete snow melting, the energy excess
that has not been used for phase changes is used to warm the underlying ground.

**2.2 New developments**

**2.2.1 New snow layering scheme**

As mentioned in Section 1, snow models of intermediate complexity are a good compromise between detailed
snow models and single-layer models. They are designed to be implemented in ESMs and, as such, should not
require excessive computational time. Although their vertical resolution is generally limited to five layers at most
(Cristea et al., 2022), several studies reported that snow models of intermediate complexity considerably improve
the representation of basic features of the snowpack and reduce biases in surface temperature when they are
compared to single-layer models (Lynch-Stieglitz, 1994; Boone and Etchevers, 2001; Dutra et al., 2012; Wang et
al., 2013). Despite these good performances, increasing the number of snow layers (with finer layers near the
surface or near the snow/ice interface) is expected to improve the modeled heat conduction within the snowpack,
the simulated temperature at the snow/ice interface, and subsequently the vertical temperature profile in the ice
and eventually the simulated SMB (Cristea et al., 2022). We therefore increased the number of snow layers from
3 to 12, following the layering scheme proposed by Decharme et al. (2016) for ISBA-ES in which the new layering
scheme is defined as:
$$\begin{cases} D_{snow}^i = min \left( \delta_i, \frac{Z_{snow}}{12} \right) \ for \ i \ \leq 5 \ or \ i \ \geq 9 \\ \\ D_{snow}^6 = 0.3 d_r - min(0, 0.3 d_r - D_{snow}^5) \\ \\ D_{snow}^7 = 0.4 d_r + min(0, 0.3 d_r - D_{snow}^5) - min(0, 0.3 d_r - D_{snow}^9) \\ \\ D_{snow}^8 = 0.3 d_r - min(0, 0.3 d_r - D_{snow}^9) \\ \\ d_r = Z_{snow} - \sum_{i=1}^5 D_{snow}^i - \sum_{i=9}^{12} D_{snow}^i \end{cases} \tag{25}$$

The $\delta_i$ values correspond to the maximum widths of the layers 1 to 5 and 9 to 12 and are fixed to $\delta_1 = 0.01$ m,
$\delta_2 = 0.05$ m, $\delta_3 = 0.15$ m, $\delta_4 = \delta_{10} = 0.5$ m, $\delta_5 = \delta_9 = 1$ m, $\delta_{11} = 0.1$ m, and $\delta_{12} = 0.02$ m. For very thin
snowpacks ($Z_{snow} \leq Z_{thin} = 0.1$ m), each layer has the same thickness $\frac{Z_{thin}}{12}$. The layer thicknesses are updated
at each time step if the first two layers (i = 1, 2) or the bottom layer (i = 12) become too thin
$\left( less \ than \ D_{snow}^i = 0.5 \times min \left( \delta_i, \frac{Z_{snow}}{12} \right) \right)$ or too thick $\left( larger \ than \ D_{snow}^i = 1.5 \times min \left( \delta_i, \frac{Z_{snow}}{12} \right) \right)$. In that
case, the snow mass and heat content are redistributed according to the new layering scheme. Otherwise, the layer
thicknesses at the current time step are kept to their previous values (i.e. at the previous time step). This allows to
maintain the density and thermal conductivity of fresh snow as long as the depth has not changed too much. This
enables the model to work more closely with more complex models in which new snow layers are associated with
a new snowfall event.

### 2.2.2 Implementation of ice layers

In case the snow mass has completely melted, ice melting occurs if the available energy is sufficient and contributes
to runoff. To account for the presence of ice below the snow layers, we implemented a new module in ORCHIDEE
to compute the heat diffusion and the vertical temperature distribution in the ice as well as the potential ice melting.
This module works in a similar way as the ES model and only accounts for vertical fluxes. The ice reservoir is
discretized into eight layers whose maximum thicknesses are fixed to 0.01, 0.05, 0.15, 0.5, 1, 5, 10 and 50 m. A
finer vertical spacing is imposed for the upper layers to better resolve heat conduction at the snow-ice or
atmosphere-ice interface. The large thickness of the bottom layer allows it to have an almost constant temperature
throughout the year as it has been observed at a few tens of meters depth (Patterson, 1994). Ice layers are only
implemented above an icy soil-type. If the icy soil is predominant in a given grid cell, then the entire surface
corresponding to this grid point will be considered as icy.
In the absence of a dynamic ice model that transports ice from the interior of the ice sheet (or glacier) to the edges,
the total ice mass may disappear entirely in the ablation zones especially in long-term simulations. To avoid such
situations, ice is considered as an infinite reservoir: melting ice contributes to runoff but, at each time step, the
amount of ice melted in the upper layers is counterbalanced by ice added at the base, and the layer thicknesses are
kept fixed to their initial value.
The vertical distribution of temperature is determined using the same numerical scheme as that for the snowpack.
If snow is still present over the ice soil, the temperature in the top ice layer is given by the temperature of the
bottom snow layer computed using Eq. (21). If snow has completely melted, the temperature in the first ice layer
is given by an expression similar to Eq. (20):
$$T_{ice}^{1} = \left[ \frac{\lambda_{ice} \cdot C_{gr\_ice} + (T_{surf} + T_{snow}^{add})}{1 + \lambda_{ice}(1 - D_{gr\_ice})} \right] \tag{26}$$
For the deeper layers, the ice temperature is expressed as follows:
$$T_{ice}^{i+1} = C_{gr\_ice} + D_{gr\_ice} \cdot T_{ice}^{i} \tag{27}$$
Similarly to the snow coefficients (see Eqs 20 and 21), $\lambda_{ice}, C_{gr\_ice}, D_{gr\_ice}$ depend on the vertical discretization
and the thermal properties of the ice. The formulations of the heat capacity ($C_{ice}$) and thermal conductivity ($\Lambda_{ice}$)
of the ice have been taken from those used in the GRISLI ice-sheet model (Yen, 1981) and are given by:
$$C_{ice} = \rho_{ice} \left( a_{ci} + b_{ci}(T_{ice} - T_0) \right) \tag{28}$$
$$\Lambda_{ice} = a_{\lambda i} exp(b_{\lambda i} \times T_0) \tag{29}$$
where $T_{ice}$ is the ice temperature, $a_{ci}$ = 2115.3 J K$^{-1}$ kg$^{-1}$, $b_{ci}$ = 7.79293 J K$^{-2}$ kg$^{-1}$, $a_{\lambda i}$ = 6.727 W m$^{-1}$ K$^{-1}$ and $b_{\lambda i}$
= -0.041 K$^{-1}$.
A major difference between the hydrology of snow and ice layers lies in the fact that ice is considered as an
impermeable medium. Hence, liquid water coming from melting ice is considered to runoff instantaneously with
no possibility of refreezing. As a result, when the ice temperature is above the melting point, the available energy
for phase change in the $i^{th}$ ice layer (J m$^{-2}$) is given by:
$$E_{ice}^i = C_{ice}^i(T_{ice}^i - T_0)D_{ice}^i \qquad (30)$$
Similarly to $S_{melt}$ (Eq. 24), the total amount of ice melt is given by:
$$M_{ice} = \frac{\sum_i E_{ice}^i}{L_f} \qquad (31)$$
and the runoff is computed as the sum of $M_{snow}$ and $M_{ice}$. Given the fact that snow drift is ignored, the surface
mass balance is computed as:
$$SMB = P_{snow} + P_{rain} - M_{snow} - M_{ice} - S_{snow} \qquad (32)$$

### 2.2.3 Other processes in the new ES model

Another modification made to the ES module concerns the inclusion of rainwater percolation within the snowpack
that may refreeze at depth as long as the maximum water holding capacity is not exceeded. In case of rain-on-
snow events, we also enhanced snow ageing by a factor of two ($f_{age} = f_{age}$ x 2). Although it sounds somewhat
arbitrary, we introduced this parameterization in the model to account for the effect of such events on
metamorphism and densification (Marshall et al., 1999), thereby lowering the albedo (Yang et al., 2023).
The snow thermal conductivity has been modified to follow a similar formulation to that used in the ISBA-ES
model (Decharme et al., 2016) and the CROCUS model (Vionnet et al., 2012) and earlier proposed by Yen (1981).
Therefore, the effective thermal conductivity in the $i^{th}$ layer now reads as:
$$\Lambda_{eff}^i = \left(a_{\lambda v} + \frac{b_{\lambda v}}{c_{\lambda v} + T_{snow}^i}\right)\frac{P_0}{P} + \Lambda_{ice}\left(\frac{\rho_s^i}{\rho_w}\right)^{1.88} \qquad (33)$$
The first term of the right-hand side that parameterizes the water vapor diffusion effects ($\Lambda_{vap}^i$) remains unchanged
(see Eq. 15). The second term replaces Eq. (14) used in the previous ES version (Wang et al. 2013) and corresponds
to the new formulation of the snow thermal conductivity ( $\Lambda_{cond}^i$). Here, the ice thermal conductivity ($\Lambda_{ice}$) differs
from the value found in Decharme et al. (2016) and is given by Eq. (29).
Besides the new snow layering scheme and the changes mentioned in this section, all the other processes simulated
in the new ES module are treated in the same way as in the three-layer version.

## 3. Experimental setup

### 3.1 Forcing by the regional atmospheric model MAR

The ORCHIDEE-ICE simulations presented in this paper were driven by the atmospheric outputs of the regional
atmospheric model MAR (Fettweis et al., 2017). This approach was motivated by the fact that MAR was initially
developed for polar regions (Gallée and Schayes, 1994). Moreover, it is coupled to a land surface scheme, SISVAT
(Soil Ice Snow Vegetation Atmosphere Transfer, De Ridder and Schayes, 1997), that includes a physically-based
snowpack model derived from the multi-layered snow model CROCUS (Brun, 1989, 1992). As such, MAR has
been extensively used to simulate the present-day climate and surface mass balance of the GrIS, and compares
well to reanalyses and available data of SMB measurements (e.g. Fettweis et al. 2017, 2020; Franco et al. 2012;
Montgomery et al. 2020; Delhasse et al., 2020). Therefore, the use of atmospheric forcings from MAR offers a
good opportunity to assess the performances of our snow model for simulating the SMB and ablation-related
processes.
The MAR simulations (1960 – 2019) were run at a 20 km x 20 km resolution. Here, we use the version v3.11.4,
identical to the version v3.11.5 for the Greenland ice sheet (Smith et al. 2023). MAR was forced every six hours
at its lateral boundaries by the meteorological fields (temperature, humidity, wind, and pressure) coming from the
ERA-40 (1960-1978, Uppala et al., 2005) and the ERA-Interim (1979-2019, Dee et al., 2011) reanalyses from the
European Centre for Medium-Range Weather Forecasts (ECMWF). Sea surface temperatures and sea ice cover,
also coming from ECMWF reanalyses, were 6-hourly prescribed.

### 3.2. The ORCHIDEE-ICE simulations

The ORCHIDEE-ICE simulations are run at a half-hourly time step with the same spatial resolution as the MAR
outputs (20 km x 20 km). The integration domain covers the whole of Greenland. ORCHIDEE-ICE is forced every
three hours by the downward shortwave and longwave radiation, the surface air temperatures and specific humidity
(all at 2 meters) and the wind speed (at 10 meters), the surface pressure and the precipitation rate (split between
rainfall and snowfall). Simulations are performed over the 1995-2019 period. The first five years (1995 to 1999)
are used for the initialization of the snowpack and are not included in the analysis. However, to obtain reasonable
thermal conditions within the ice layers, a longer time integration is required. Thus, we performed a preliminary
spin-up experiment over the same 25 years to infer an initial vertical temperature profile for the subsequent
ORCHIDEE-ICE simulations.
The name and the characteristics of the different experiments presented in this paper are summarized in Table A1.
Using the experimental design described above, we first ran the ES model with three and twelve snow layers (STD-
3L and STD-12L experiments respectively) to evaluate the added value of the new layering scheme. These
experiments were carried out with the albedo parameters used in the CMIP6 ORCHIDEE version (Chéruy et al.,
2020) and referred hereafter to as the standard snow albedo parameters.
Due to the strong sensitivity of the SMB to the albedo, we also conducted two additional experiments with
modified values of the albedo parameters. In the ASIM-12L experiment, we used the parameters inferred from the
approach of Raoult et al. (2023). This latter was based on a data assimilation experiment using the MODIS
retrievals. The main goal of their study was to optimise the albedo parameters so as to improve the albedo for the
ice sheet as a whole, while giving an extra weight to the edges where the greatest amount of runoff is produced.
In doing this, they also succeeded in improving the model-data fit between the ORCHIDEE albedo and MODIS
retrievals over the whole GrIS, and reducing the root-mean-square error (RMSE) by ~25 %. However, their work
was done with a previous version of the ORCHIDEE-ICE model with only three snow layers and in which the ice
layers were not implemented. Instead, ice was mimicked by a soil type whose porosity and volumetric water
content were set to 98% to simulate a soil filled with frozen water.
The logical follow-up to the work of Raoult et al. (2023) would have been to apply the optimisation algorithm to
the new version of ORCHIDEE-ICE. Since this approach is highly time-consuming, it has not yet been carried
out, albeit it will be the focus of further investigations. Therefore, using the new ORCHIDEE-ICE model version,
we adopted a manual tuning approach (i.e. trial-and-error method) to adjust the albedo parameters (OPT-12L
experiment). This procedure consists in 1/ changing the parameter values, the new value being taken from the

range reported in Table 1, 2/ running the model with the new parameter values, 3/ evaluating the model performance (in terms of SMB and its components) using statistical criteria (e.g. RMSE between MAR and ORCHIDEE-ICE) and 4/ repeating steps 1/ to 3/ until an acceptable calibration is obtained (i.e. acceptable values of SMB, runoff, refreezing and sublimation).

Finally, to assess the impact of the climatic fields used as inputs of ORCHIDEE-ICE, we performed another experiment (ERA5-12L experiment) by forcing the model with the ERA5 reanalysis (Hersbach et al., 2020) and using the same albedo parameters than in OPT-12L experiment.

**Table 1:** List of the ORCHIDEE-ICE experiments (first column) with values chosen for the different albedo parameters (standard albedo parameters for STD-3L and STD-12L, optimized albedo parameters inferred from Raoult et al. (2023) for ASIM-12L and manual-tuned parameters for OPT-12L and ERA-12L. Values in brackets indicate for each parameter the range of values considered in the manual tuning approach.

| Exp. | Nb of snow layers | $A_{aged}$ [0.50 - 0.70] | $B_{dec}$ [0.10 - 0.40] | $\tau_{dec}$ [1.0 - 10.0] | $\delta_c$ [0.2 - 2.0] | $\omega_1$ [1.0 - 7.0] | $\omega2$ [0.5 - 6.0] | $\tau_{max}$ [40 - 60] | $\alpha_{ice}$ [0.30 - 0.50] |
|---|---|---|---|---|---|---|---|---|---|
| STD-3L | 3 | 0.620 | 0.170 | 10 | 0.2 | 7 | 4 | 50 | 0.400 |
| STD-12L | 12 | 0.620 | 0.170 | 10 | 0.2 | 7 | 4 | 50 | 0.400 |
| ASIM-12L | 12 | 0.553 | 0.320 | 6.911 | 0.783 | 3.037 | 3.974 | 56.183 | 0.476 |
| **OPT-12L** | **12** | **0.580** | **0.280** | **2.0** | **1.0** | **3** | **6** | **54** | **0.420** |
| ERA-12L | 12 | 0.580 | 0.280 | 2.0 | 1.0 | 3 | 6 | 54 | 0.420 |

**4. Methodology for the model performance evaluation**

**4.1 Comparison with MAR outputs**

Our first objective is to assess the performance of the ORCHIDEE ICE model in representing the GrIS SMB. The period under study spans over the 2000-2019 period. As mentioned in Section 3, MAR has revealed good capabilities in simulating the SMB of present-day Greenland when compared to observational data. Therefore, at the scale of the entire GrIS, our evaluation is made with respect to the MAR outputs (Figs 2a-5a). In all simulations presented in this paper except ERA5-12L, the forcing fields of the ORCHIDEE-ICE model are provided by MAR outputs. These include solid and liquid precipitation which constitute the accumulation (and the climatic) component of the SMB. By using the MAR forcing, our analysis of the ability of ORCHIDEE-ICE to reproduce ablation processes (runoff and sublimation) is made easier and is not biased by the use of another forcing.

**4.2 Comparison with available data**

In this study, we compared the albedo computed in ORCHIDEE-ICE with satellite-derived estimates of daily albedo. We used Collection 6 from the MOD10A1 product (Hall et al., 1995) retrieved from the NASA space-borne sensor MODIS. We chose this product because it has a good spatiotemporal coverage over snow-covered areas. It is also one of the best performing products in terms of comparison with in situ data (Urraca et al., 2022, 2023). Moreover, while studies based on the previous Collection 5 reported deficiencies at latitudes higher than 70°N (Alexander et al., 2014), substantial improvements have been made to Collection 6 by using all available

observations for the acquisition period against only four observations per day in Collection 5 (https://lpdaac.usgs.gov/products/mcd43d11v006/, last access 01/22/2024). As a result, better quality retrievals are obtained at high latitudes despite a slight negative bias (Urraca et al., 2022). To avoid inaccuracies in retrieved data due to the presence of clouds or aircraft condensation trails, the MOD10A1 albedo product used in this study was further processed by Box et al. (2017): data have been de-noised, gap-filled, corrected for the sun-angle bias and validated using daily ground albedo values from the PROMICE (Programme for Monitoring of the Greenland ice sheet, Fausto et al., 2021) and GC-net automatic weather stations (Box et al. 2017).

We aggregated the albedo data (500 m x 500 m) onto the MAR grid to make the comparison between MODIS data and the ORCHIDEE-ICE outputs. In this study, we used the albedo data covering the 2000-2017 period because data for the years 2018 and 2019 were undefined. The resulting dataset may be used to calibrate the mean ORCHIDEE-ICE albedo, computed as the mean between the visible (from 0.4 to 0.7 μm) and near infrared (from 0.7 to 2.5 μm) bands (see Section 2).

As in Fettweis et al. (2020), we also evaluated the modelled SMB with the Machguth et al. (2016) SMB database. Daily outputs are used here over 2000-2019. Modelled SMB were linearly interpolated to the measurement point location and corrected for the elevation difference between the MAR native topography at 20km and the one provided in the SMB database. This was done by using a space-varying SMB–elevation gradients, as proposed by Franco et al. (2012) and Noël et al. (2016). Finally, measurements not included in the 2000–2019 period and records located outside the 20km MAR ice mask are discarded from the evaluation.

**4.3 Statistical metrics**

To evaluate our model performances, we used statistical metrics:

The root-mean-square error (RMSE) has been computed using the monthly mean variables averaged over 2000-2019 for the SMB and its components, and over 2000-2017 for the albedo. It is defined as:

$$RMSE = \sqrt{\frac{1}{N}\sum_{i=1}^{N}(X_{OR}(i) - X_{MAR}(i))^2} \qquad (34)$$

where $X_{OR}(i)$ and $X_{MAR}(i)$ represent the ORCHIDEE-ICE and the MAR variables respectively at each grid point $i$, N is the number of unmasked grid points (i.e. related only to the ice-covered area) and $i$ stands for the $i^{th}$ grid point. The RMSE is a metric widely used to compare different models but it has some shortcomings related to the fact that higher weights are given to larger errors. We there used additional statistical criteria to provide a more in-depth picture of our analysis. We computed the spatial RMSE (SRMSE) which gives a measure of the quadratic difference averaged over time between values simulated by both models over the entire GrIS domain and at each time step. Thus, by taking the temporal variations in the simulated time series into account, the spatial RMSE makes it possible to assess the model's performance both over the entire geographical domain and over the time period under consideration. It is computed as follows:

$$SRMSE = \sqrt{\frac{1}{N_t \times N}\sum_{t=1}^{N_t}\sum_{i=1}^{N}\left(X_{i,OR}(t) - X_{i,MAR}(t)\right)^2} \qquad (35)$$

$X_{i,OR}(t)$ and $X_{i,MAR}(t)$ are respectively the ORCHIDEE-ICE and MAR variables at each grid point $i$ and each time
step $t$. $N_t$ is the number of time steps. In contrast to the RMSE, we used the daily simulated values to compute the
SRMSE.
While the RMSE and SRMSE give an indication of the magnitude of the absolute difference between both models,
it is also important to calculate the area-weighted average bias (hereafter, areal-mean bias) of each grid point in
order to examine whether the model variables simulated by ORCHIDEE-ICE are underestimated (negative bias)
or overestimated (positive bias). This bias (MB) is given by:
$$MB = \frac{\sum_{i=1}^{N} A_i (X_{OR}(i) - X_{MAR}(i))}{\sum_{i=1}^{N} A_i} \qquad\qquad (36)$$
where $A_i$ is the surface area of each grid point.
Finally, we also examined the probability density functions (PDFs) and performed a Cramer-von Mises (CVM)
test (Anderson, 1962) to compare the MAR and ORCHIDEE-ICE distributions of a given variable. The CVM test
integrates the quadratic differences between the two models over the whole distributions (including the tails of the
distributions). In this sense, it is more powerful and more sensitive to departures from the reference distribution
(i.e. MAR) than the widely used Kolmogorov-Smirnov test (Stephens, 1970), which is based on the absolute value
of the greatest distance between the two distributions.
**5. Results**
**5.1 Evaluation against MAR for standard albedo parameters**
Figures 2 to 4 display the spatial distribution of runoff, sublimation and refreezing simulated by MAR (panels a)
and by ORCHIDEE-ICE in the STD-3L (panels b) and STD-12L (panels c) experiments. The main runoff areas
simulated with MAR are located on the western edge albeit, to some extent, runoff occurs in all peripheral areas
of the ice sheet (Fig. 2a). Locations of the ablation zones are well represented in ORCHIDEE-ICE but are limited
to a very narrow band, especially in the STD-3L simulation (Fig. 2b). Increasing the number of snow layers favors
the inland expansion of the ablation areas on the western and northern margins (Fig. 2c). However, this expansion
remains too restricted compared to MAR (Fig. 2a). In the ablation areas, differences in the amount of runoff exceed
1.5 mm day$^{-1}$ (Fig. S1). Integrated over the whole ice sheet (Table 2), the runoff values computed in STD-3L (152
Gt yr$^{-1}$) and STD-12L (205 Gt yr$^{-1}$) experiments for the 2000-2019 period are respectively 59 % and 45 % lower
compared to MAR (375 Gt yr$^{-1}$). As a consequence of the considerably smaller amount of runoff in ORCHIDEE-
ICE, and thus of surface meltwater, refreezing is also much lower (Table 2) and less extended (Figs. 3a-c)
compared to MAR. It can be noted, however, that the disagreement is less pronounced with the STD-12L
experiment (Fig. S2), which underlines the benefit of increasing the number of snow layers.

Runoff 2000-2019 (mm day$^{-1}$)

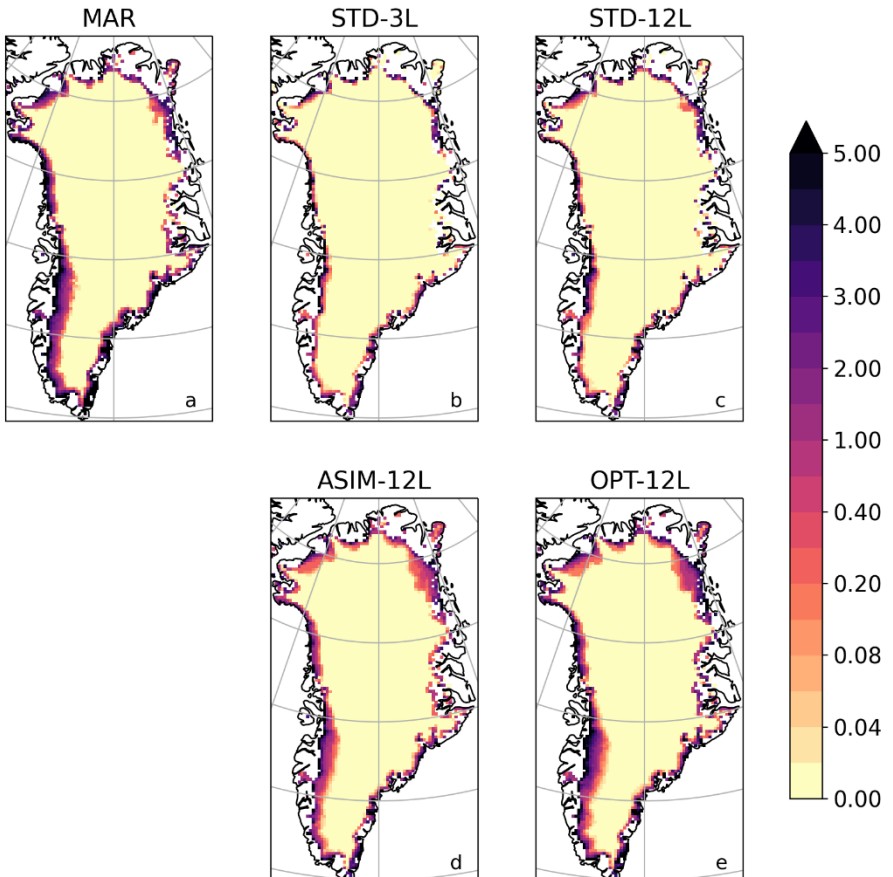

**Figure 2:** Spatial distribution of the runoff (in mm day$^{1}$) averaged over the 2000-2019 period and simulated with MAR (a) and the ORCHIDEE-ICE model (b-e) using: the three-layer snow scheme and the standard albedo parameters (b), the twelve-layer snow scheme and the standard albedo parameters (c), the twelve-layer snow scheme and the albedo parameters optimised using a data assimilation technique (Raoult et al., 2023) and a previous version of the ORCHIDEE-ICE model (d), the twelve-layer snow scheme and the albedo parameters obtained after manual tuning (e).

**Table 2:** Simulated values of SMB, runoff, sublimation and refreezing integrated over the entire Greenland ice
sheet and averaged over the 2000-2019 period (2nd column). Evaluation of simulated SMB and SMB components
is done with respect to MAR outputs using values of root-mean-square error (3rd column), areal mean bias and
(4th column) and spatial root-mean-square error (5th column).

| Experiments | SMB (Gt yr$^{-1}$) | RMSE (in mm day$^{-1}$) | Areal mean bias (in mm day$^{-1}$) | Spatial RMSE (in mm day$^{-1}$) |
|---|---|---|---|---|
| **MAR** | **286** | | | |
| STD-3L | 504 | 0.976 | 0.351 | 3.050 |
| STD-12L | 450 | 0.786 | 0.264 | 2.809 |
| ASIM-12L | 466 | 0.706 | 0.290 | 2.602 |
| **OPT-12L** | **301** | **0.464** | **0.024** | **2.530** |
| ERA5-12L | 352 | | | |
| Experiments | Runoff (Gt yr$^{-1}$) | RMSE (in mm day$^{-1}$) | Areal mean bias (in mm day-1) | Spatial RMSE (in mm day$^{-1}$) |
| **MAR** | **375** | | | |
| STD-3L | 152 | 1.107 | - 0.357 | 3.157 |
| STD-12L | 205 | 0.922 | - 0.272 | 2.900 |
| ASIM-12L | 217 | 0.829 | - 0.254 | 2.639 |
| **OPT-12L** | **336** | **0.592** | **-0.063** | **2.539** |
| ERA5-12L | 273 | | | |
| Experiments | Sublimation (Gt yr$^{-1}$) | RMSE (in mm day$^{-1}$) | Areal mean bias (in mm day$^{-1}$) | Spatial RMSE (in mm day$^{-1}$) |
| **MAR** | **82** | | | |
| STD-3L | 32 | 1.000 | - 0.081 | 0.200 |
| STD-12L | 33 | 0.096 | - 0.079 | 0.203 |
| ASIM-12L | 5 | 0.134 | - 0.124 | 0.226 |
| OPT-12L | 52 | 0.077 | - 0.049 | 0.274 |
| ERA5-12L | 89 | | | |
| Experiments | Refreezing (Gt yr$^{-1}$) | RMSE (in mm day$^{-1}$) | Areal mean bias (in mm day-1) | Spatial RMSE (in mm day$^{-1}$) |
| **MAR** | **186** | | | |
| STD-3L | 72 | 0.336 | - 0.183 | 1.254 |
| STD-12L | 104 | 0.269 | -0.131 | 1.134 |
| ASIM-12L | 90 | 0.313 | - 0.155 | 1.182 |
| **OPT-12-L** | **158** | **0.240** | **-0.046** | **1.316** |
| ERA5-12L | | | | |




Refreezing 2000-2019 (mm day$^{-1}$)

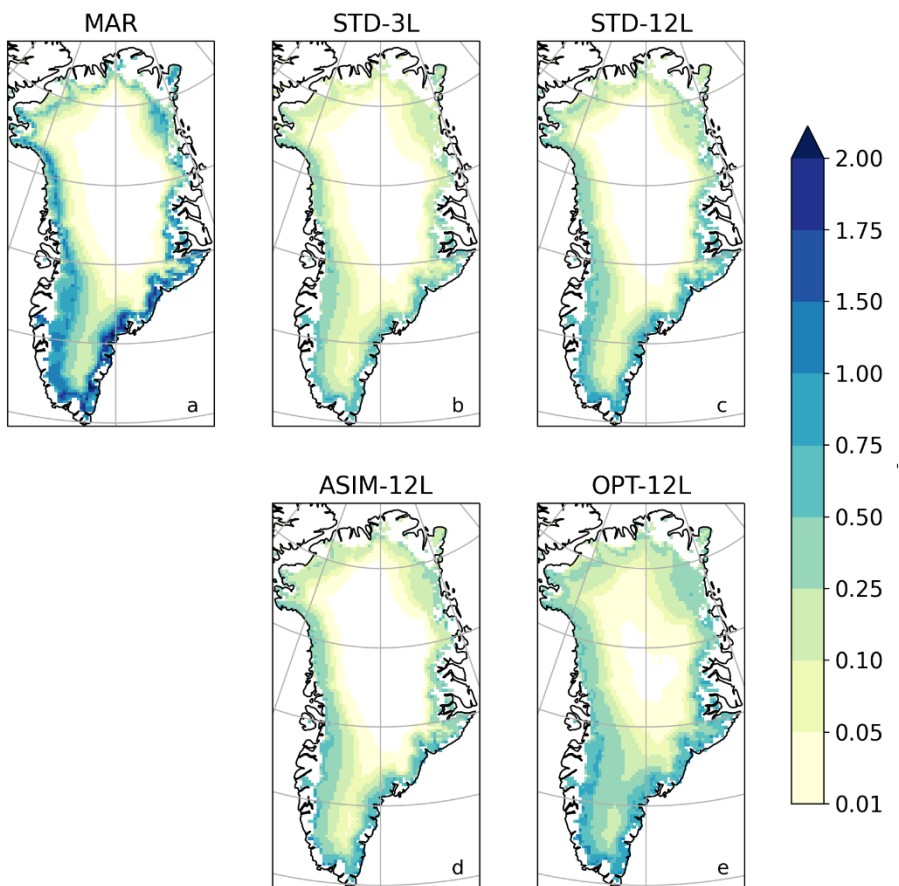


**Figure 3:** Same as Figure 2 for the simulated refreezing (in mm day$^{-1}$).
Large differences between MAR and ORCHIDEE-ICE also arise regarding sublimation (32 and 33 Gt yr$^{-1}$ in the
STD-3L and STD-12L experiments respectively, against 82 Gt yr$^{-1}$ for the 2000-2019 period in MAR). This feature
concerns the entire ice sheet but is even more striking in peripheral areas (Figs 4 and S3). In central Greenland,
differences are smaller, but ORCHIDEE-ICE simulates a little condensation (Fig. 4) whereas MAR does not.
The differences in simulated runoff and in sublimation between MAR and ORCHIDEE-ICE translate into
overestimated SMB values simulated with ORCHIDEE-ICE (504 and 450 Gt yr$^{-1}$ in STD-3L and STD-12L against
286 Gt yr$^{-1}$ in MAR; see also Figs. 5 and S4). Since inland regions are dominated by the accumulation signal,
which is provided by the MAR outputs, the SMB anomalies are primarily driven by differences in the ablation
components occurring at the edges of the ice sheet, and exceed 2 mm day$^{-1}$ in most parts of the western and
southeastern margins.
An important conclusion that can be drawn from these results is that the use of a better resolved snow layering
scheme (twelve-layer as opposed to a three-layer snow scheme) reduces the mismatch between MAR and
ORCHIDEE-ICE. This is mainly illustrated by the integrated SMB and runoff values which are respectively ~35%
lower and ~11% higher in STD-12L, translating into reductions of RMSE values (~19% and ~17% for SMB and
runoff respectively, see Table2), areal mean bias (~25% and ~24% respectively), and, to a lesser extent, of the
spatial RMSE (~8% for both SMB and runoff). Nevertheless, the differences with MAR are still too large for the
model to be used as a reliable tool to compute the GrIS SMB.

Sublimation 2000-2019 (mm day$^{-1}$)

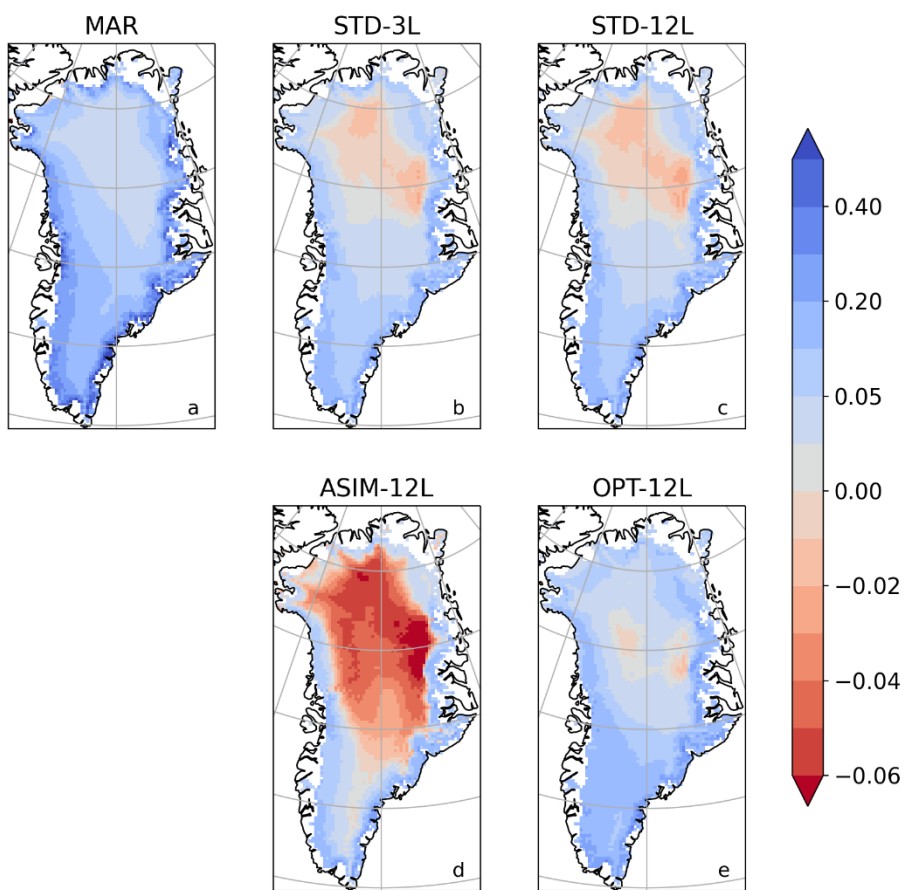


**Figure 4:** Same as Figure 2 for the simulated sublimation (in mm day$^{-1}$). Negative values indicate condensation.

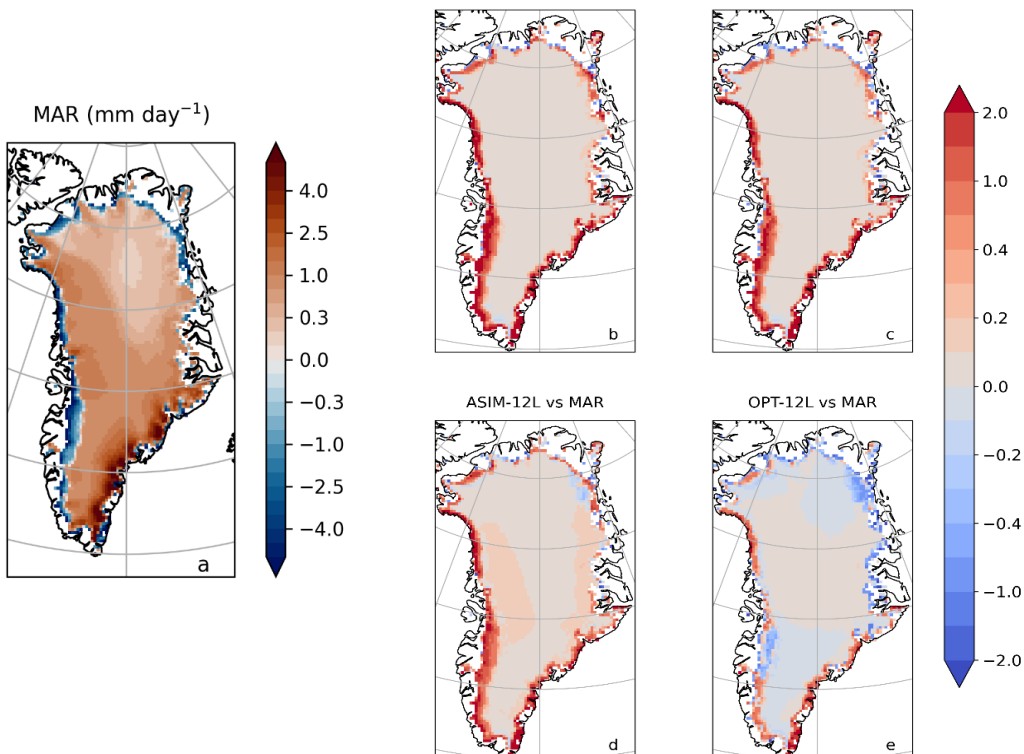


**Figure 5:** Spatial distribution of the GrIS SMB simulated with MAR (in mm day$^{-1}$) and averaged over the 2000-2019 period (a) Differences in the GrIS surface mass balance between MAR and the ORCHIDEE-ICE model (b-e) with the standard parameter values of the albedo parameterisation and the three-snow layering scheme (b). Panels (c-e) correspond to simulations performed with the updated twelve-snow layering scheme for standard values of the albedo parameters (c), optimised values of the albedo parameters (d), values of the albedo parameters obtained after manual tuning (e).

## 5.2. SMB and runoff for modified albedo parameters

### 5.2.1 Impact of optimised albedo parameters

As snow is a highly reflective medium, little changes in albedo may produce large changes in the surface energy balance, and thus, in the SMB. In the GrIS interior, there is generally a quite good agreement between the summer albedo computed by MAR and the standard ORCHIDEE-ICE simulations (i.e. STD-3L and STD12-L experiments, Figs. 6b and 6c and S5) with slight negative anomalies of less than 0.05. Negative anomalies (~ -0.1) also appear, mainly in the northern part of the ice sheet, but with only little consequences on surface melting owing to the very cold conditions in this region. However, on the western margin, where most of the melting takes place, larger snow albedo values are found in ORCHIDEE-ICE. This leads to underestimated surface temperatures compared to MAR (Fig. 7) and, thus, to undervalued runoff that may explain part of the discrepancies between MAR and ORCHIDEE-ICE. There are also differences between the observations provided by MODIS retrievals and the MAR albedo (Figs. 6a and 6f), especially in the northern and southern parts, and the western margin. On the other hand, the summer albedo computed in the STD-3L and STD-12L experiments (Figs. 6g and 6h) are generally too low in the interior of the ice sheet, and too high on the western margin with differences from 0.05 to 0.15.

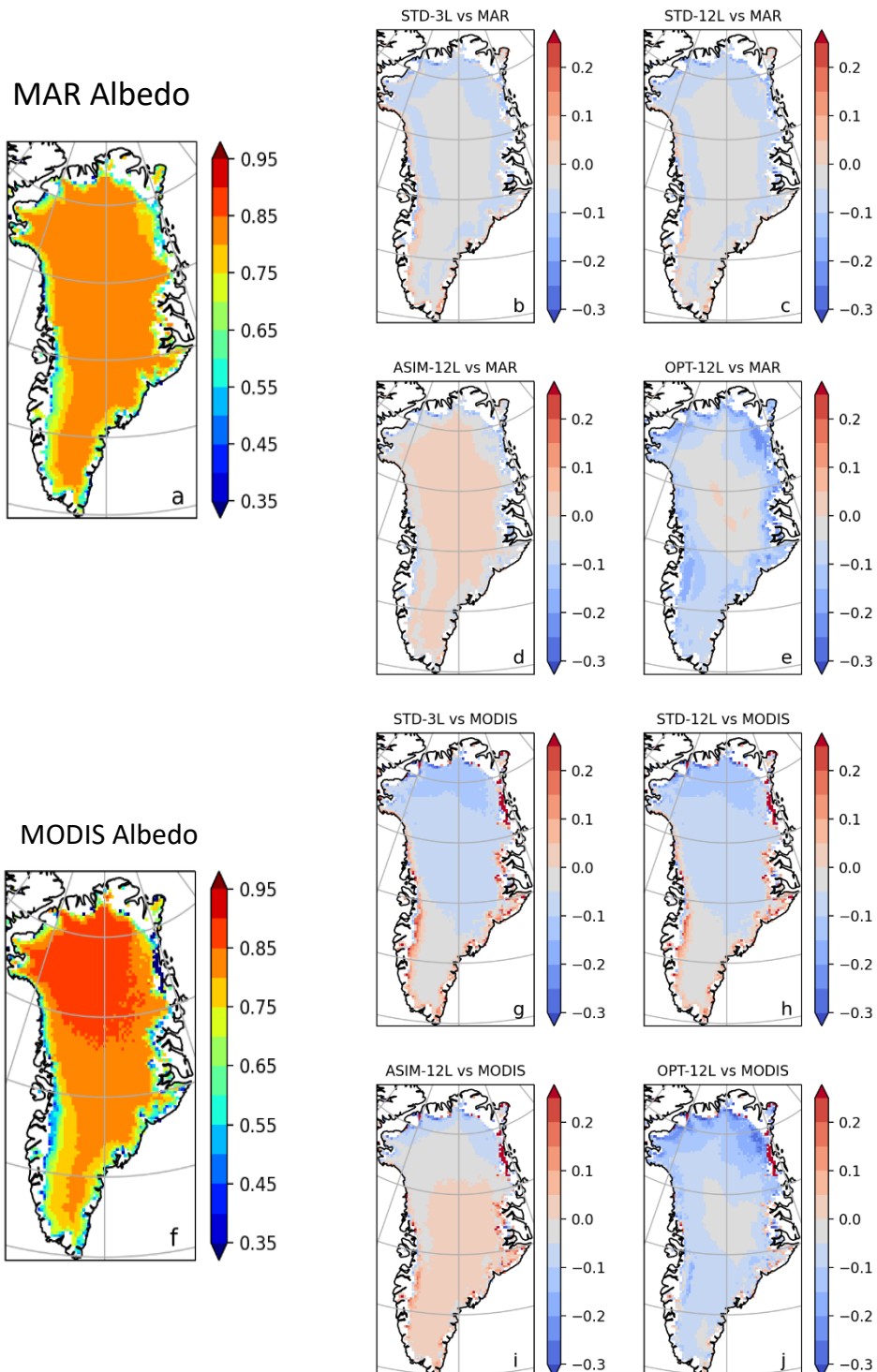

**Figure 6:** Left: Spatial distribution of the summer (JJA) albedo computed with MAR (a) and MODIS (f) and averaged over the 2000-2017 period. Right: Differences between the albedo computed with ORCHIDEE-ICE and MAR (b,c,d,e) and between ORCHIDEE-ICE and MODIS (g,h,i,j) for the three-layer snow scheme and the standard albedo parameters (b,g), the twelve-layer snow scheme and the standard albedo parameters (c,h), the albedo parameters inferred from a data assimilation technique and using a previous version of the ORCHIDEE-ICE model (d,i), the albedo parameters obtained after manual tuning (e,j).

As mentioned in Section 3.2, we investigated the sensitivity of the SMB and its components to the albedo. We first performed an ORCHIDEE-ICE experiment (ASIM-12L) with the optimised albedo parameters inferred from Raoult et al. (2023). Figure 6i illustrates how the representation of the albedo has been improved in the ASIM-12L experiment compared to STD-12L (Figs. 6h, S5 and S8). Model-data discrepancies are now reduced with differences lower than 0.05 except in the northernmost parts of the ice sheet. The RMSE decreased by ~26% (Table 3), which is quite consistent with Raoult et al. (2023). The ablation areas are now better represented (Fig. 2d) due to increased surface temperatures (Fig. 7c) as a result of lower albedo values on the western margin (Fig. 6i).

**Table 3:** Albedo RMSE values (2nd column), areal mean biases (3rd column) and spatial RMSE with respect to MODIS (top) and MAR (bottom).

| Experiments | RMSE (w.r.t MODIS) | Areal mean bias (w.r.t MODIS) | Spatial RMSE (w.r.t MODIS) |
|---|---|---|---|
| MAR | 0.076 | - 0.005 | |
| STD-3L | 0.098 | - 0.047 | 0.098 |
| STD-12L | 0.097 | - 0.051 | 0.097 |
| ASIM-12L | 0.072 | 0.001 | 0.072 |
| OPT-12L | 0.111 | - 0.008 | 0.092 |
| Experiments | RMSE (w.r.t MAR) | Areal mean bias (w.r.t. MAR) | Spatial RMSE (w.r.t MAR) |
| STD-3L | 0.055 | - 0.042 | 0.055 |
| STD-12L | 0.058 | - 0.047 | 0.058 |
| ASIM-12L | 0.051 | 0.006 | 0.040 |
| OPT-12L | 0.092 | - 0.047 | 0.092 |

However, despite the smaller mismatch between modeled ASIM-12L albedo and MODIS retrievals and the better representation of the ablation areas, the simulated amount of runoff (217 Gt yr$^{-1}$) integrated over the whole GrIS has been only slightly improved with respect to STD-12L (Figs. 2d) and remains quite different from MAR outputs (Figs. 2a). In addition, the simulated SMB (466 Gt yr$^{-1}$) has even been slightly degraded (Figs. 5a and 5d) due to negative temperature anomalies in central Greenland extending until the southern tip (Fig. 7c) resulting from slightly higher albedo values compared to MAR and MODIS (Figs 6a, 6i).

Summer snow surface temperature differences with MAR

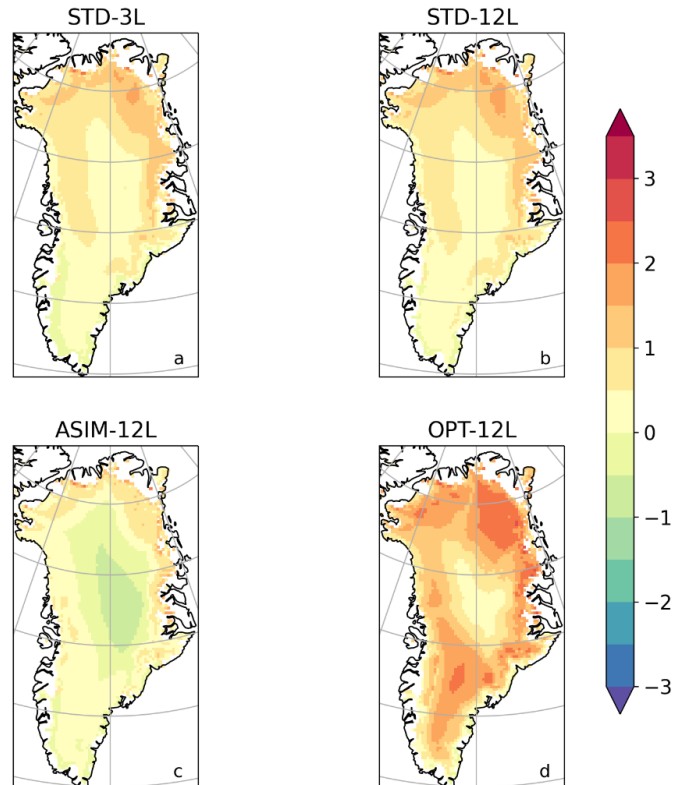

**Figure 7:** Spatial distribution of the snow temperature differences with respect to MAR averaged over the 2000-2019 period (in °C) simulated for the STD-3L (a), STD-12L (b), ASIM-12L (c) and OPT-12L (d) experiments.

The low performance for the SMB computation in ASIM-12L is not solely due to a small amount of runoff but also to strong negative values of sublimation (i.e. large condensation) over central Greenland (Fig. 3d) resulting in an average level of - 5 Gt yr$^{-1}$ over the entire ice sheet compared to 82 Gt yr$^{-1}$ in MAR (Table 2). In the ASIM-12L experiment, the albedo in the central GrIS region is slightly higher (up to 0.05) than the albedo retrieved from MODIS (Fig. 6i), while the albedo computed with MAR is slightly lower (Figs. 6a and 6f). This explains why the ASIM-12L surface temperatures are smaller than those simulated with MAR. This can lead, therefore, to lower saturation pressures that can drop below the dew point and thus produce solid condensation. This result highlights the key influence of the albedo on surface processes and, in particular, illustrates how a small departure from observations may lead to strong biases in sublimation estimates.

**5.2.2 Manual tuning**

As mentioned in Section 3, we have not yet performed a data assimilation experiment to calibrate the new twelve-layer ES model, given the computational cost of such an experiment. Instead, we chose to follow a trial-and-error approach. As runoff dominates the SMB signal, our primary objective was to improve the runoff computation by reducing the summer albedo values in the main ablation areas (i.e. the western margin). Given the number of albedo parameters, several options are available to achieve this:

- lowering the albedo of aged snow ($A_{aged}$) and/or the albedo of fresh snow ($A_{aged} + B_{dec}$);
- modifying the parameter controlling the decay rate of snow albedo ($\tau_{dec}$);

- increasing snow age by changing the parameters related to snow aging: the minimum snowfall thickness to reset snow age to zero ($\delta_c$), the tuning parameters $\omega_1$, $\omega_2$ (see Eq. 10) and the maximum snow age ($\tau_{max}$);

- changing the ice albedo ($\alpha_{ice}$) because it can also affect SMB and runoff computation if the snowpack melts entirely during summer months in some places and give rise to bare ice.

Owing to the various influences of the albedo parameters, we had to find a compromise so as to lower the albedo in ablation areas and improve the computation of runoff and SMB, while keeping reasonable albedo values in the GrIS interior. Among the values we tested for each of the parameters, the set of parameters providing the best agreement with MAR outputs (for SMB and SMB components) is highlighted in bold in Table 1 (OPT-12L experiment). Compared to the ASIM-12L experiment (Figs. 6i, S5, S8), the albedo mismatch between ORCHIDEE-ICE (OPT-12L experiment) and MODIS is amplified, especially along the western margin and in the northern sector with differences reaching 0.25 and 0.3 respectively (Fig. 6j). Nevertheless, these results were expected since our manual tuning was designed to increase the magnitude of the ablation components (especially runoff) and to decrease the SMB, and therefore to lower the albedo values with a direct impact on surface temperatures, hence surface melting and sublimation.

### 5.2.3 Impact on SMB components

Using the new set of albedo parameters obtained with the manual tuning approach, the ablation areas are now much more extended than those simulated in the STD-12L experiment (Figs. 2c and 2e). Compared to MAR (Fig. 2a), they are even wider in the northern part due to increased surface temperatures (Fig. 7d) in response to lower albedo values (up to -0.25). The total amount of runoff averaged over the 2000-2019 period is now 336 Gt yr$^{-1}$ (against 375 Gt yr$^{-1}$ in MAR). For the OPT-12L experiment, the RMSE value decreased by ~40% compared to STD-12L (Table 2). In the same way, the sublimation (52 Gt yr$^{-1}$) and refreezing (158 Gt yr$^{-1}$) better match with MAR (Table 2). In particular, condensation over central Greenland has been considerably reduced, notably with respect to ASIM-12L, but sublimation is still underestimated along the GrIS edges and in the southern part (Fig. 4e). The increase in refreezing (with respect to STD-12L and ASIM-12L) in the GrIS interior (Fig. 3e) is likely linked to lower summer albedo values (Figs. 6e and 6j) leading to a smaller amount of melting compensated by refreezing. In the main ablation areas, a larger refreezing is produced and thus a better agreement with MAR, though still insufficient, is obtained.

These results for the SMB components are evidently associated with an improved representation of the SMB itself (Fig. 5e) which now reaches 301 Gt yr$^{-1}$ (286 Gt yr$^{-1}$ obtained with MAR. Indeed, the RMSE and the spatial RMSE values have been reduced by ~41% and 10% respectively for the SMB (~28% and 9% for the runoff) compared to the STD-12L experiment (Table 2). An even more striking result concerns the areal mean bias which has been lowered by one order of magnitude. These improvements are also illustrated in Figure 8, which displays the monthly mean values for each grid point of the SMB components simulated with ORCHIDEE-ICE as a function of the same MAR variables (see for example the correlation coefficient for both SMB and runoff for the OPT-12L experiment). However, our results are less conclusive for sublimation and refreezing. Although, the areal-mean bias and the RMSE values indicate a better match between the OPT-12L and the MAR simulations, the spatial RMSE values are greater compared to the three other ORCHIDEE-ICE experiments, suggesting a lower temporal consistency between OPT-12L and MAR. In addition, the correlation coefficients for sublimation and refreezing

are also smaller (Fig. 8). On the other hand, the best overlaps between the probability density functions between MAR and the ORCHIDEE-ICE experiments is undoubtedly obtained for OPT-12L, as shown in Figs. S6-S7 and the scores of the CVM tests reported in Table S1.

**Figure 8:** Representation of the simulated SMB (1st row), runoff (2nd row), sublimation (3rd row) and refreezing (4th row) simulated with ORCHIDEE-ICE as a function of the same MAR variables: STD-3L (1st column), STD-12L (2nd column), ASIM-12L (3rd column) and OPT-12L (4th column). The different points represent the monthly mean values over the period 2000-2019 for each of the grid points. The regression line is displayed in red (R is the regression coefficient) and the line y = x is in black.

Despite these encouraging results, it is important to underline that the improved SMB simulation in OPT-12L is
achieved through the albedo reduction, and therefore, to some extent, come from error compensation. However,
the reduced albedo also makes it possible to compensate for the effect of some missing mechanisms, such as the
lack of consideration of snow-atmosphere interactions or the absence of an explicit representation of snow
metamorphism, which has a direct impact on the density profile, the albedo itself and the temperature profile.

## 5.3 Vertical temperature and density profiles

To go a step further and gain a better understanding of the above results, it is also important to explore the internal
processes of the snowpack. To achieve this, we chose to focus on the vertical temperature and density profiles.
Figure 9 depicts the snow temperatures simulated ORCHIDEE -ICE as a function of the MAR snow temperatures
at 20 cm and 1 m depth of the snowpack. These plots show that the temperatures simulated in STD-3L, STD-12L
and OPT-12L behave approximately in the same way when compared to those of MAR. In the first 20 cm,
ORCHIDEE-ICE is slightly warmer than MAR for temperatures between -30°C and -10°C, despite a few slightly
colder grid points appearing in the range of -20°C to -10°C. The ASIM-12L experiment presents the best agreement
with MAR, although slightly lower temperatures. These features reflect directly the behavior of surface
temperatures (Fig. 7) that strongly influence the upper snowpack layers. Another key point arising from these plots
is the very good agreement between MAR and ORCHIDEE-ICE for temperatures above -10°C. This suggests that
the potential runoff that could occur in the first tens of centimeters of the snowpack should not be so much affected.
However, the departure from MAR increases with snow depth, especially for low temperatures. For example, at
1 m depth, differences of 3-4°C are obtained (Fig. 9) and may exceed 5°C for deeper levels (not shown). These
enhanced differences with MAR are likely due to a positive feedback related to the thermal conductivity (see Eq.
33): As snow temperature increases by 1°C in a given layer, the thermal conductivity increases by one order of
magnitude.
As pointed out by Domine et al. (2019), the snow thermal regime and snow density are strongly coupled. As an
example, they mentioned the work of Fréville (2015) who showed that an error of 1°C in the surface temperature
can lead to errors on snow density of 100 kg m$^{-3}$. Our experiments show that for a depth of 20 cm, the higher the
surface temperature, the lower the snow density on average (Fig. 10). On the other hand, in the ASIM-12L
experiment, snow temperatures are lower, compared to the three other ORCHIDEE-ICE experiments, and snow
densities are larger. This contradicts a number of studies (e.g. Kojima, 1967; Anderson 1976, Mizukami and Perica,
2008), which have shown that in a warmer snowpack, snow grains become rounded and are more prone to be
compacted more easily, hence leading to an increase in snow density. However, in our model this process cannot
be reproduced as snow metamorphism is only accounted for through snow ageing. Conversely, in deeper layers,
the model is more effective at densifying (Fig. 10), in line with the fact that warmer snow becomes more plastic
and compacts more easily. In particular; between 20 cm and 1 m depth, the RMSE computed between OPT-12L
and MAR has been reduced from 79.63 kg m$^{-3}$ to 30.22 kg m$^{-3}$. Beyond 600 kg m$^{-3}$; the ORCHIDEE-ICE densities
are generally below those of MAR because the maximum density is fixed to 750 kg m$^{-3}$ (see Section 2). However,
the comparison of our results on snow density with those of MAR should be viewed with caution because, to the
best of our knowledge, the snow density simulated by MAR has not been evaluated against available observations.

Snow Temperature 2000-2019 (°C)

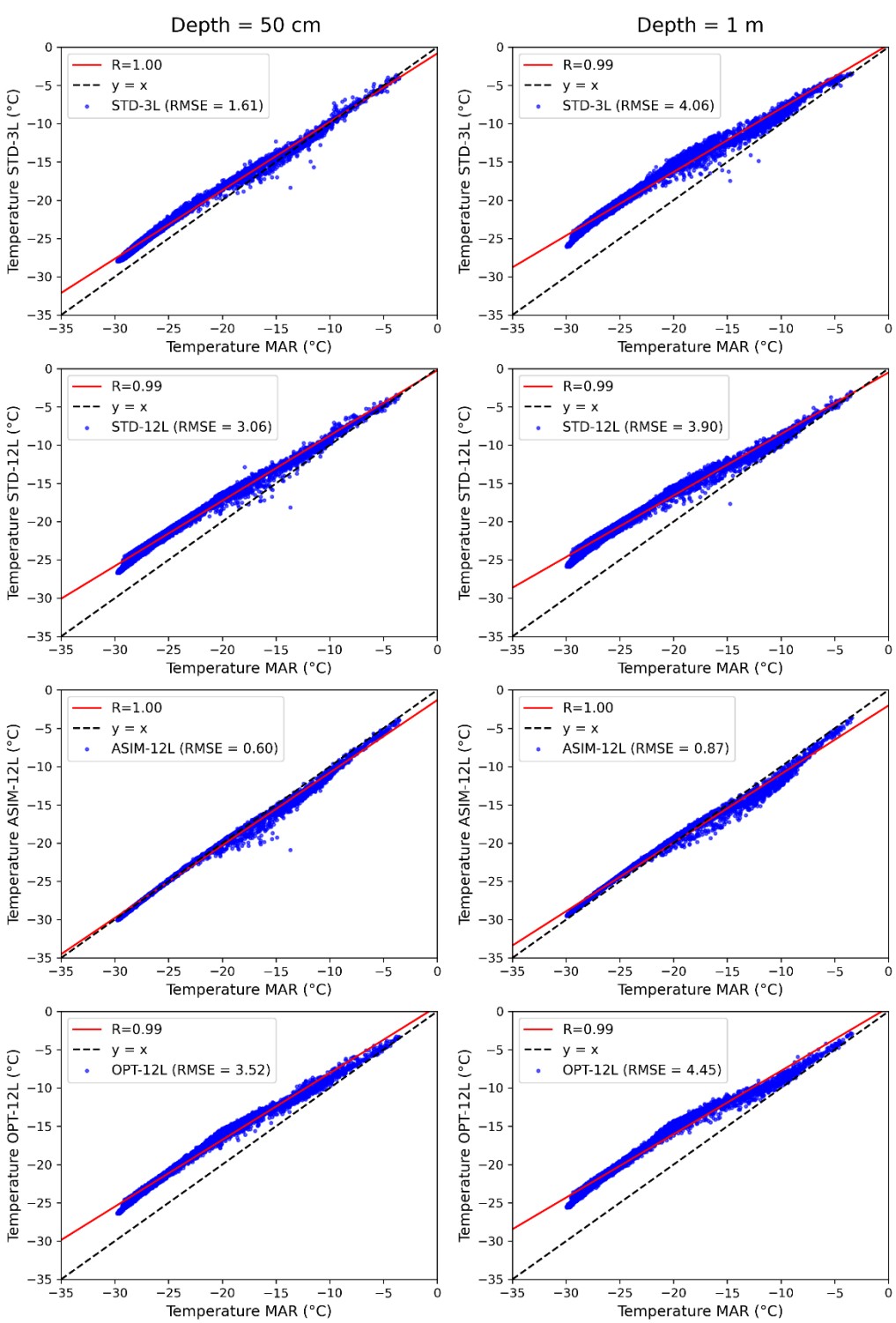


**Figure 9:** Representation of the ORCHIDEE-ICE simulated snow temperatures at 50 cm (left) and one-meter depth (right) as a function of the MAR snow temperatures. The different points represent the monthly mean values over the period 2000-2019 for each grid point. The regression line is displayed in red (R is the regression coefficient) and the line y = x is in black.


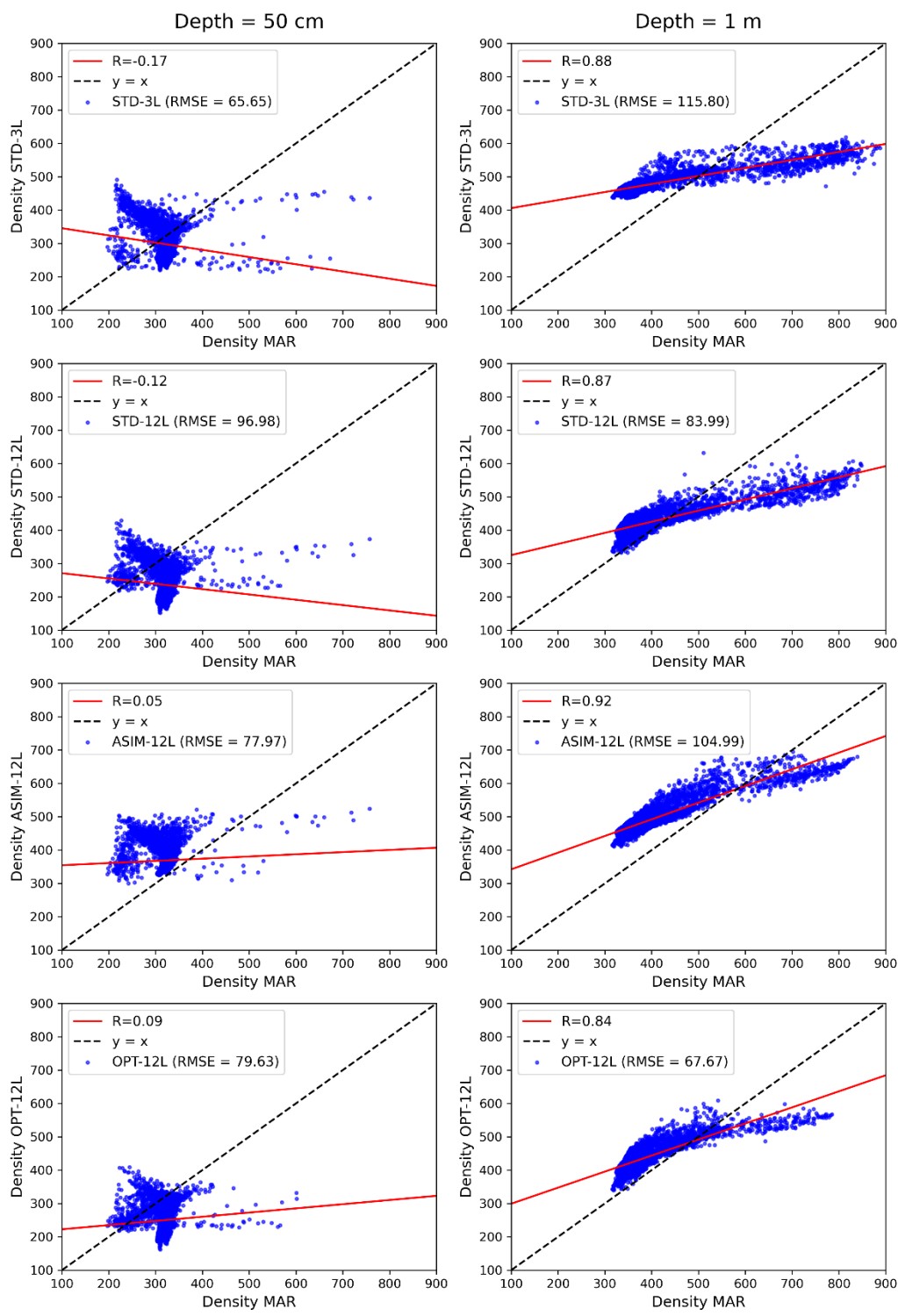


**Figure 10:** Same as Figure 9 for snow density



**5.4 SMB evolution: impact of the climate forcing**

The results presented in the previous sections were averaged over the 2000-2019 period (for SMB and the SMB components) and over the 2000-2017 period (for the albedo). In this part, we present the temporal evolution of the SMB between the years 2000 and 2019 (Fig. 11). Figure 11 shows that whatever the ORCHIDEE-ICE experiment under consideration, the evolution of the yearly integrated SMB is in accordance with the evolution simulated by the MAR model. In particular, the years in which extreme melting events were recorded (such as 2012 and 2019) are perfectly well represented (Bennartz et al. 2013; Tedesco and Fettweis 2020). As expected, the best agreement with MAR is obtained for the OPT-12L experiment as a result of the calibration of the albedo parameters. When forced by the ERA-5 meteorological fields, and using the manually-tuned parameters, ORCHIDEE-ICE simulates higher SMB values and a lower runoff (Fig. 11 and Table 2), especially during the first period of the time series (2000-2008). However, the evolution of the yearly integrated SMB in the ERA5-12L experiment follows exactly the same interannual variations as for the OPT-12L experiment forced with MAR (Fig. 11). This indicates that the surface climate simulated by MAR is close to that derived from the ERA-5 products. Moreover, in a comparative study of the ERA-5 reanalyses, Arctic System reanalysis and MAR performances, Delhasse et al. (2020) showed that MAR outperforms ERA-5 for the near-surface temperatures when compared to observations from automatic weather stations. As the surface melt, and thus the SMB, largely depend on near-surface temperatures, there is, therefore, a strong interest in using MAR to force our snow model and to compare its performances to those of MAR.

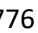

**Figure 11:** Evolution of the yearly surface mass balance of the Greenland ice sheet simulated with MAR (black), ORCHIDEE-ICE forced by MAR outputs (STD-3L and STD-12L: yellow, solid and dashed lines respectively; OPT-12L: red line), ORCHIDEE-ICE forced by the ERA-5 reanalyses (green line).

In this paper, we have so far limited the comparison of our results to those of MAR. However, as mentioned in
Section 4, we also evaluated the simulated SMB with 353 daily SMB observations from the PROMICE database
available over the 2000-2019 period (Machguth et al., 2016; Mankoff et al., 2021). In addition, it is also interesting
to evaluate our model results against observations when ORCHIDEE-ICE is forced by climatic fields independent
from MAR outputs. To address this issue, we plotted the modelled SMB for OPT-12L, ERA5-12L and MAR for
the grid points located closest to the observation sites as a function of the PROMICE SMB measurements (Fig. 12).
We also provided statistical elements for the comparison between MAR, the five ORCHIDEE-ICE
experiments and the SMB observations (Table 4). This model-data comparison confirms the conclusions we
reached when evaluating the performance of our model against MAR outputs, namely the significant improvement
in our results when moving from STD-3L to OPT-12L. Moreover, although the bias between the OPT-12L SMB
and the observed SMB is twice as high as for MAR, the model-data correlation is of the same order of magnitude
as for MAR (Table 4).

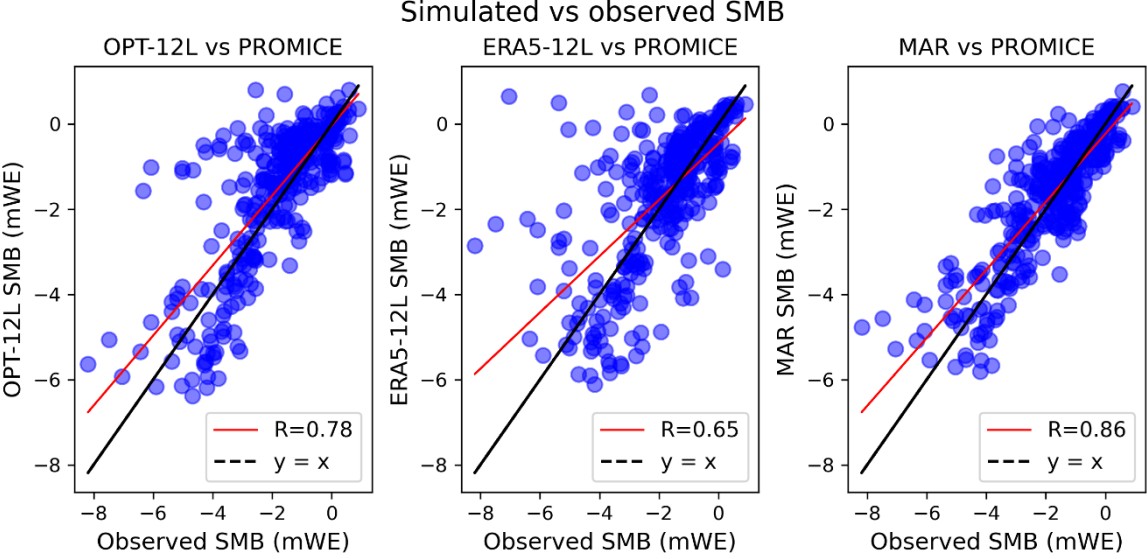

**Figure 12:** Simulated SMB in the OPT-12L experiment and in MAR as a function of the observed SMB from the
PROMICE network. As the observed SMB values are not all available over the same time interval, the
measurements are given in meter water equivalent (mWE). 353 observations were available over the 2000-2019
period. Each simulated SMB value corresponds to the grid points located closest to the observation sites. The red
line is the regression line with R being the correlation coefficient and the dashed black line indicates the line y =
x.
The ERA5-12L experiment also produces a good agreement with the observations. Despite a lower correlation
coefficient than for MAR and OPT-12L, the mean bias is of the same order of magnitude as that of MAR and the
RMSE on the SMB obtained is the lowest for any of the experiments. It is clear that the SMB simulated in the
experiments forced by MAR is partly driven by the climate simulated by MAR itself (for the accumulation
component). However, the results obtained with ERA5-12L clearly show that the behaviour of our model is
consistent whatever the climate forcing used. Nevertheless, it should be reminded that the resolution of
ORCHIDEE-ICE corresponds to that of the model used as a forcing. For ERA5-12L, the resolution is about twice
as fine as for the experiments forced by MAR (20 km x 20 km). Thus, to make our comparison between ERA5-
12L, MAR and/or OPT-12L more robust, we should have used MAR with a resolution of 10 km x 10 km. It cannot
therefore be ruled out that the results for OPT-12L would then have provided a better comparison with the
PROMICE data than ERA5-12L.
**Table 4:** Comparison of the simulated SMB in MAR, STD–3L, STD-12L, ASIM-12L and OPT-12L with the
SMB observations from the PROMICE network. The bias is computed as the average between modelled and
observed SMB for each grid point. Note that the values of the bias and the RMSE are given in mWE as the observed
SMB values are not all available over the same time interval.

| Experiments | Bias (mWE) | Correlation | RMSE (mWE) |
| --- | --- | --- | --- |
| MAR | 0.14 | 0.86 | 0.82 |
| STD-3L | 0.94 | 0.67 | 1.70 |
| STD-12L | 0.68 | 0.73 | 1.43 |
| ASIM-12L | 0.74 | 0.75 | 1.33 |
| OPT-12L | 0.30 | 0.78 | 1.13 |
| ERA5-12L | 0.17 | 0.65 | 1.07 |

**6. Discussion and concluding remarks**
The land surface component of the IPSL ESM used for CMIP6 included a three-layer snowpack model operating
over continental surfaces. However, this snow scheme was not adapted to glaciated surfaces, which is a major
drawback and makes it impossible to compute the surface mass balance over ice sheets or glaciers. The aim of this
paper was therefore to present the new developments made to adapt the snow model to ice-covered areas and to
document its performance. Our first step was to calibrate the snow albedo parameterisation over the Greenland ice
sheet. To have a set of climate variables covering the whole ice sheet, we chose to force our model by the
atmospheric outputs of the MAR regional model which shows very good performances to simulate the surface
climate and thus offers undeniable advantages for the representation of the physical processes related to snow and
ice, in particular surface melting (Delhasse et al., 2020). We have shown that the ablation-related processes are
highly dependent on the choice of the albedo parameters. The set of parameters obtained after manual tuning (OPT-
12L experiment) provides a good agreement between the SMB computed in ORCHIDEE-ICE and MAR.
However, as outlined in Section 5.2.3, this improvement is mainly the result of albedo lowering. The summer
albedo computed with this set of parameters has been degraded compared to MAR and MODIS and to the albedo
computed in the ASIM-12L experiment (based on the MODIS-optimised albedo parameters) as shown in Table 3
and in Figures 6i-6j and S5, S8. While the RMSEs computed between ORCHIDEE-ICE and MAR for SMB and
runoff have been reduced by ~39% and ~33% respectively from ASIM-12L to OPT-12L, the RMSE for albedo
has increased by 47% (Table 3). The mismatch between MODIS retrievals and OPT-12L albedo is mainly observed
in the northernmost part of the ice sheet and, to a lesser extent, on the western edge.
A more objective method would have been to perform a data assimilation experiment similar to the one presented
in Raoult et al. (2023) using the new version of the ORCHIDEE-ICE model. However, albedo is not the only
important parameter governing the snowpack evolution. The albedo parameters inferred from Raoult et al. (2023)'s
optimisation greatly improve the representation of the albedo, but degrade the other model outputs compared to
those obtained with the manually-tuned albedo parameters. This is most likely because their optimisation overfits
the albedo retrievals without applying constraints to the other processes strongly impacting the SMB components
and controlling the state of the snowpack (e.g. snow compaction, snow density, snow viscosity). This supports the
recommendation for a multi-objective optimisation using not only albedo data, but also vertical temperature and
density profiles as well as SMB observations. Since this type of approach is highly time-consuming, it has not yet
been undertaken but could be the objective of a future study.
However, the reduction in albedo in the current ORCHIDEE-ICE version can compensate for missing processes.
For example, snow drift, transmission of solar radiation, or the effect of light absorbing particles on the albedo are
ignored. Metamorphism is not explicitly represented although its effect on the albedo and the vertical density
profile are accounted for (albeit in a crude manner) through the snow ageing function $f_{age}$ (Eq. 7) and the $\psi_{snow}$
function (Eq. 17) respectively.
In the GrIS, snow erosion has often been considered as a second-order component of mass loss in ablation areas
compared to melt water. However, in the ice sheet interior, sublimation and snow erosion are dominant processes
in removing mass from the surface, and may have, therefore, a significant impact on SMB (van Angelen et al.,

853  2011).

Taking into account the transmission of solar radiation within the snowpack can lead to a warming of the internal
layers, with higher temperatures near the surface and lower temperatures at depth due to the exponential decrease
in heat transfer. This results in a temperature gradient that influences the metamorphism of snow grains and thus
accelerates densification (Colbeck, 1983). We showed that the ORCHIDEE-ICE temperatures inside the snowpack
were higher than those simulated by the MAR model. A likely hypothesis to explain this behaviour relies on the
reduction in albedo, which leads to excessively high surface temperatures. However, it is important to note that
heat transfer can promote snow melting, which in turn can percolate at depth and refreeze, affecting both the runoff
and the vertical structure of the snowpack through changes in density (Colbeck, 1983). Quantifying all these
processes requires, therefore, the proper representation of solar absorption, which is itself strongly dependent on
snow optical properties (Warren, 1982) and, therefore, on snow grain size (Libois et al., 2013). Since
metamorphism is not explicitly represented in the model, we assumed that representing solar absorption was not a
priority in our modeling approach, even if this choice is debatable. However, in the near future, a more
sophisticated albedo scheme based on a transfer radiative model accounting for light-absorbing particles and snow
grain size (Kokhannovsky and Zege, 2004) will be implemented in the ORCHIDEE-ICE model. This will allow
to represent the backward and forward scattering processes as well as light absorption.
In addition, there are also structural deficiencies related to the fact that in ORCHIDEE-ICE, a single energy balance
is computed in one grid cell. This is detrimental for the albedo computation especially at the edges of the ice sheet
where several surface types may coexist in a 20 km x 20 km mesh. However, the implementation of a multi-tile
energy balance is currently under development.
Finally, as our simulations have been run in off-line mode, the snow feedback onto the atmosphere has not been
taken into account, contrary to the MAR model fully coupled to a snow scheme derived from CROCUS (Brun,
1989, 1992). Ignoring snow-atmosphere feedback may potentially lead to biases related to surface processes and
to an improper representation of the energy and humidity flux exchanges at the snow-atmosphere interface. For
example, forcing our model with the atmospheric temperature at 2m derived from the full coupled MAR simulation
could lead to an underestimation of the energy available at the snow-atmosphere interface, resulting in less
snowmelt compared to what is simulated in coupled mode. However, our manual tuning approach aims at limiting
the potential underestimation of the surface meltwater production. Conversely, any potential bias in the MAR
forcing may also affect our results (Dietrich et al., 2024). To overcome this problem, it would have been interesting
to force ORCHIDEE-ICE by meteorological fields recorded at the automatic weather stations. This has not been
done in this study because the meteorological fields required to force ORCHIDEE-ICE were not all available at
the PROMICE stations and because our first objective was to obtain a reasonable estimate of the SMB and its
components at the scale of the entire GrIS.
Despite the potential improvements that could still be made to ORCHIDEE-ICE to enhance the model's
performance, the developments presented in this paper represent a major step forward. Indeed, they now allow the
ice-sheet surfaces to be handled by the land surface model, consistently with all the other surface types, and not
by the atmospheric component of the IPSL model (LMDZ), as was the case up to now. In addition, the new snow
model can now be applied to the continental glaciers replacing the very crude snow scheme used previously. Our
developments enable us to provide a reasonable estimate of the surface mass balance of the Greenland ice sheet,
in very good agreement with that simulated by the MAR model which was used as a reference in this study. These
developments constitute a first step towards the full coupling between the IPSL global climate model and ice-sheet
models.

 **Appendix A:**

 **Table A1:** List of variables used in ORCHIDEE-ICE and related to snowpack and ice processes

| Symbol | Variable | Units | Value/Range |
|---|---|---|---|
| $\alpha$ | Surface albedo of the grid cell | | |
| $\alpha_{snow}$ | Albedo of a snow-covered surface | | |
| $\alpha_{snow-free}$ | Albedo of snow-free surface | | |
| $\alpha_{ice}$ | Ice albedo | | |
| $\delta_c$ | Snowfall thickness necessary for resetting the snow age to zero | kg m$^{-2}$ s$^{-1}$ | |
| $\eta_{snow}$ | Snow viscosity | Pa s | |
| $\eta_0$ | Snow viscosity parameter | Pa s | $3.7 \times 10^7$ |
| $\Lambda_{snow} (\Lambda_{ice})$ | Snow (ice) thermal conductivity | W m$^{-1}$ K$^{-1}$ | |
| $\Lambda_{eff} = \Lambda_{snow}$ | Effective snow thermal conductivity | W m$^{-1}$ K$^{-1}$ | |
| $\Lambda_{cond}$ | Snow thermal conductivity | W m$^{-1}$ K$^{-1}$ | |
| $\Lambda_{vap}$ | Snow thermal conductivity | W m$^{-1}$ K$^{-1}$ | |
| $\lambda_{snow}$ | Integration coefficient for snow thermal profile numerical scheme | | |
| $\lambda_{ice}$ | Integration coefficient for ice thermal profile numerical scheme | | |
| $\rho_{snow}$ | Snow density | kg m$^{-3}$ | 917 |
| $\rho_{ice}$ | Ice density | kg m$^{-3}$ | |
| $\rho_{water}$ | Water density | Kg m$^{-3}$ | 1000 |
| $\rho_{air}$ | Air density | kg m$^{-3}$ | |
| $\rho_t$ | Parameter of the maximum water holding capacity | kg m$^{-3}$ | 200 |
| $\rho_\psi$ | Parameter for the effect of metamorphism in the snow density | kg m$^{-3}$ | 150 |
| $\sigma_{snow}^i$ | Pressure of the snow load over the i$^{th}$ layer | Pa | |
| $\tau_{snow}$ | Snow age | days | |
| $\tau_{dec}$ | Time constant of the albedo decay | days | |
| $\tau_{max}$ | Maximum snow age | days | |
| $\omega_1, \omega_2$ | Tuning constants for snow albedo | | |
| $A_{aged}$ | Snow albedo of old snow | | |

| | | | |
|---|---|---|---|
| $A_i$ | Surface area of the $i^{th}$ grid point | m² | |
| $a_\eta$ | Snow viscosity parameter | K⁻¹ | 8.1 x 10⁻² |
| $a_\psi$ | Parameter for the effect of metamorphism | s⁻¹ | 2.8 x 10⁻⁶ |
| $a_\lambda$ | Parameter for snow thermal conductivity | W m⁻¹ K⁻¹ | 0.02 |
| $a_{\lambda v}$ | Parameter of snow thermal conductivity from vapor transport | W m⁻¹K⁻¹ | -0.06023 |
| $a_{ci}$ | Parameter of heat capacity of the ice | J K⁻¹ kg⁻¹ | 2115.3 |
| $a_{\lambda i}$ | Parameter of ice thermal conductivity | W m⁻¹ K⁻¹ | 6.627 |
| $B_{dec}$ | Decay rate of snow albedo | | |
| $b_\eta$ | Snow viscosity parameter | m³ kg⁻¹ | 1.8 x 10⁻² |
| $b_\psi$ | Parameter for the effect of metamorphism | K⁻¹ | 4.2 x 10⁻² |
| $b_\lambda$ | Parameter of snow thermal conductivity | W m⁵ K⁻¹ kg⁻² | 2.5 10⁻⁶ |
| $b_{\lambda v}$ | Parameter of snow thermal conductivity from vapor transport | W m⁻¹ | -2.5425 |
| $b_{ci}$ | Parameter of heat capacity of the ice | J K⁻² kg⁻¹ | 7.79293 |
| $b_{\lambda i}$ | Parameter of ice thermal conductivity | K⁻¹ | -0.041 |
| $c_\psi$ | Parameter for the effect of metamorphism | m³kg⁻¹ | 460 m³kg⁻¹ |
| $c_{\lambda v}$ | Parameter of snow thermal conductivity from vapor transport | K | -289.99 |
| $C_{soil}$ | Surface heat capacity of soil | J m⁻² K⁻¹ | |
| $C_{snow}$ | Snow heat capacity | J m⁻² K⁻¹ | |
| $C_{snow}^v, (C_{ice})$ | Snow (ice) volumetric heat capacity | J m⁻³ K⁻¹ | |
| $C_{gr\_snow}, D_{gr\_snow}$ | Integration coefficients for snow thermal profile numerical scheme | | |
| $C_{gr\_ice}, D_{gr\_ice}$ | Integration coefficients for ice thermal profile numerical scheme | | |
| $D_{snow}^i$ | Depth of the i$^{th}$ snow layer | m | |
| $D_{lwe}^i$ | Snow water equivalent in the i$^{th}$ snow layer | m | |
| $D_{ice}^i$ | Depth of the i$^{th}$ ice layer | m | |
| $dt$ | ORCHIDEE time step | s | 1800 |
| $E_{snow}^i (E_{ice}^i)$ | Energy available to induce phase changes in the snowpack (in the ice) | W m⁻² s⁻¹ | |
| $F_C$ | Heat conductive flux | W m⁻² | |

| | | | |
|---|---|---|---|
| $f_{age}$ | Snow age function | | |
| $G_{snow}$ | Surface energy flux over snow-covered areas | W m$^{-2}$ | |
| $G_{surf}$ | Surface energy flux | W m$^{-2}$ | |
| $H$ | Sensible heat flux | W m$^{-2}$ | |
| $H^i_{snow}$ | Heat content in the i$^{th}$ snow layer | W m$^{-2}$ s$^{-1}$ | |
| $H_{rainfall}$ | Heat release from rainfall | W m$^{-2}$ | |
| $LE$ | Latent heat flux | W m$^{-2}$ | |
| $L_s$ | Latent heat of sublimation | J kg$^{-1}$ | 2.8345 10$^6$ |
| $L_f$ | Latent heat of fusion | J kg$^{-1}$ | 333.7 |
| $LW_{net}$ | Net longwave radiation | W m$^{-2}$ | |
| $M_{snow}$ ($M_{ice}$) | Total amount of snow (ice) melt at each time step | kg m$^{-2}$s$^{-1}$ | |
| $N$ | Number of unmasked grid points over the entire Greenland ice-covered area | | |
| $N_t$ | Number of daily time steps over the years 2000-2019 | | |
| $P$ | Atmospheric pressure | hPa | |
| $P_0$ | Reference pressure | hPa | 1000 |
| $P_{snow}$ | Snowfall amount during the time step dt | kg m$^{-2}$s$^{-1}$ | |
| $P_{rain}$ | Rainfall amount during the time step dt | kg m$^{-2}$s$^{-1}$ | |
| $Q_{air}$ | Air specific humidity at 2 m | - | |
| $Q_{sat}$ | Saturated specific humidity at 2 m | - | |
| $q_{cdrag}$ | Transfer coefficient | - | |
| $r_{min}$ | Parameter of the maximum water holding capacity | | 0.03 |
| $r_{max}$ | Parameter of the maximum water holding capacity | | 0.10 |
| $SCF$ | Snow cover fraction | - | |
| $S_{snow}$ | Snow sublimation | kg m$^{-2}$ s$^{-1}$ | |
| $SMB$ | Surface mass balance | kg m$^{-2}$s$^{-1}$ | |
| $SW_{net}$ | Net shortwave radiation | W m$^{-2}$ | |
| $T_{air}$ | Surface air temperature at 2 m | K | |
| $T_{soil}$ | Surface temperature | K | |

| | | | |
|---|---|---|---|
| $T_0$ | Freezing temperature | K | 273.15 |
| $T_{snow}^{add}$ | Snow temperature adjustment | K | |
| $T_{snow}$ $(T_{ice})$ | Snow (ice) temperature | K | |
| $U$ | Wind speed at 10 m | m s$^{-1}$ | |
| $W_{liq}^i$ | Liquid content in the i$^{th}$ snow layer | m | |
| $W_{max}^i$ | Maximum water holding capacity of the i$^{th}$ snow layer | m | |

***Code availability:*** The source code for the ORCHIDEE-ICE version used in this study is freely available online via the following address https://doi.org/10.14768/d82899b4-09b4-4337-abb1-75886602fe72 (IPSL Data Catalogue, 2024). The ORCHIDEE model code is written in Fortran 90 and is maintained and developed under a subversion (SVN) control system at the Institut Pierre Simon Laplace (IPSL) in France.

***Data availability:*** The MAR outputs are available at ftp://ftp.climato.be/fettweis (last access 30 October 2020). The MODIS Greenland albedo retrievals MOD10A1 are available at https://doi.org/10.22008/FK2/6JAQPK (last access 22 January 2024, Box et al., 2022). The surface mass balance observations from the PROMICE network are available at https://dataverse.geus.dk/dataverse/PROMICE (last access 06/10/2024, Machguth et al., 2016; Mankoff et al., 2021).

***Author contributions:*** SC conceived the project funding the study. SC, CD, FM and CO co-designed the research and contributed to the code developments. SC and CD performed the preliminary tests with strong support from FM and CO. CD implemented the new snow-layering scheme and the new icy soil type. XF ran the MARv3.11.4 simulations, provided the MAR outputs and performed the comparison between the simulated SMB and the PROMICE dataset. NR provided the albedo parameters obtained from the data assimilation experiment. SC, CD, FM and CO analysed the results with contributions from NR and XF. SC wrote the original draft, with contributions from CD, FM and CO, and generated the figures. SC and PC analysed the vertical temperature and density profiles. All co-authors provided comments on the manuscript.

***Competing interests:*** The authors declare that one of the co-authors is a member of the editorial board of *The Cryosphere*.

***Acknowledgements:*** This study has received funding from Agence Nationale de la Recherche - France 2030 as part of the PEPR TRACCS programme under grant numbers ANR-22-EXTR-0010 and ANR-22-EXTR-0008. The work has also been supported by the French INSU/LEFE OSCAR project. The authors would like to thank all members of the SNOW working group gathering members from the Institut Pierre Simon Laplace (IPSL, France) and the Institut des Géosciences de l'Environnement (IGE, France) for numerous and fruitful discussions. They also thank J.-Y. Peterschmitt for technical support and the core ORCHIDEE team for maintaining the model and especially J. Ghattas for helping merge the ORCHIDEE-ICE code into the trunk version of the model. Data from the Programme for Monitoring of the Greenland Ice Sheet (PROMICE) are provided by the Geological Survey of Denmark and Greenland (GEUS) at http://www.promice.dk. They include sites financially supported by the Glaciobasis programme as part of Greenland Ecosystem Monitoring (https://g-e-m.dk/), maintained by GEUS (ZAK, LYN) and by Asiaq Greenland Survey (NUK_K). The WEG stations are paid for and maintained by the University of Graz. The authors are very grateful to two anonymous reviewers who provided insightful comments that greatly help to improve the manuscript, and to the editor Marie Dumont.

932

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
