# Peer review of "Modelling snowpack on ice surfaces with the ORCHIDEE land surface"

_EGUsphere, 2024_

## Referee Comment (RC1)

**General comments**

This paper consists in essence of two parts. First, it provides a model description, documenting the Explicit Snow (ES) scheme implemented in the version of the ORCHIDEE land surface model used in the IPCL-CM Earth System Model for IPCC AR6, and the changes made to apply the ES scheme to glaciers and ice sheets, also including an increase in its vertical resolution. Second, the enhanced version of the model, named ORCHIDEE-ICE, is applied to the computation of snow mass balance (SMB) and its components over the Greenland ice sheet, in experiments forced with input data from the regional MAR model. With default parameter settings, the model clearly overestimates the SMB compared to MAR, and the improvement of vertical resolution reduces this bias only slightly. Good agreement with the MAR results for the SMB components is achieved by tuning the snow albedo parameterization, but at the cost of underestimated surface albedo over Greenland.

Overall, this paper paves the way towards including ice-sheet surfaces in the land surface module of the IPSL-CM ESM. At the same time, it provides a cautionary story, as satisfactory simulation of SMB is only achieved when the snow surface albedo is biased low.

I think this paper is a useful contribution to the topic, and it is generally well written. Therefore I recommend publication of the paper in The Cryosphere subject to the relatively minor corrections/clarifications listed below.

**Specific comments**

1. lines 78–80: I think it would be worth mentioning, as an example of a relatively sophisticated GCM snow scheme, the Community Land Model (Lawrence et al. 2019) which includes the SNICAR scheme (Flanner and Zender 2005; 2006) and is employed at least in CESM and NorESM. This modelling system simulates prognostically snow density, grain size, liquid water and absorbing aerosols in a multi-layer snowpack, and computes snow albedo and absorption of solar radiation within the snowpack based on snow grain size and aerosol concentrations. Recently, SNICAR has been extented to also compute the albedo of glacier ice (Whicker et al. 2022), although (to my knowledge) it has not yet been coupled with a glacier model.

REFERENCES:

Flanner, M. G., and C. S. Zender (2005), Snowpack radiative heating: Influence on Tibetan Plateau climate, Geophys. Res. Lett., 32, L06501, doi:10.1029/2004GL022076.

Flanner, M. G., and C. S. Zender (2006), Linking snowpack microphysics and albedo evolution, J. Geophys. Res., 111, D12208, doi:10.1029/2005JD006834.

Lawrence, D. M., Fisher, R. A., Koven, C. D., Oleson, K. W., Swenson, S. C., Bonan, G., et al. (2019). The Community Land Model version 5: Description of new features, benchmarking, and impact of forcing uncertainty. Journal of Advances in Modeling Earth Systems, 11, 4245–4287. https://doi.org/10.1029/2018MS001583

Whicker, C. A., Flanner, M. G., Dang, C., Zender, C. S., Cook, J. M., and Gardner, A. S.: SNICAR-ADv4: a physically based radiative transfer model to represent the spectral albedo of glacier ice, The Cryosphere, 16, 1197–1220, https://doi.org/10.5194/tc-16-1197-2022, 2022.

2. lines 162–163. I'm puzzled about the sign convention here. Firstly, what is actually meant by saying that "$G_{surf}$" is computed negatively"? That $G_{surf}$ is negative when the net energy flux is directed downwards, and positive when it is directed upwards? Second, judging by Eqs. (3) and (4), $H_S$ and $H_L$ are positive upwards. So if $H_S$ and $H_L$ increase, more energy is directed away from the surface… which means that according to Eq. (1), $G_{surf}$ becomes more negative, which implies (according to line 163) that there is more warming of the surface?! This does not sound physically correct. Please, check and explain carefully the sign convention used in Eqs. (1–4). A graph showing which terms are positive upwards or downwards could be helpful.

3. Line 210: "$\delta_c$ is the critical value of solid precipitation necessary for resetting the snow age to zero". This is only roughly true. For very cold temperatures, setting $P = \delta_c$ results in $\tau_{snow}$ reduced by a factor of $e$, while for warm temperatures it may actually become negative.

4. Section 2.1.2: What is assumed about the vertical distribution of solar radiation absorbed by snow? And in ice (Section 2.2.2)? I'm getting the impression that all energy is deposited on the surface, and it is then distributed in snow only through diffusion, but it would be helpful to state this explicitly.

5. line 303: (Eq. 21): Does this indeed mean that the temperature of a snow layer only depends on the temperature of the layer above, and not on the temperature of the layer below? I would expect heat diffusion to work in both directions. The same question goes for ice (Eq. 27).

6. lines 394–395: "we also enhanced snow ageing by a factor of two in case of a rainfall event". This sounds rather ad-hoc, and furthermore, it is not clear what it actually means. I think you should report, with an equation, how rainfall impacts snow age ($\tau_{snow}$) in the model.

7. lines 444–445: "Reducing the root-mean-square error (RMSE) by $\sim 22\%$." Which quantity are you referring to? The RMSE in albedo? Also, Rauolt et al. (2023) reported this number to be "over 25%".

8. line 455: Which parameter(s) were the target in the optimization? Surface albedo or something else (SMB, runoff?). This seems to be said on line 605, but it should be reported already here.

9. lines 534–535: This should be the other way round: "$\sim 11\%$ lower and $\sim 35\%$ higher".

10. lines 585–586. This sentence is not very clear. Does the implementation of ice layers make runoff smaller or larger?

11. lines 690: I would suggest to conclude this paragraph by saying explicitly (e.g.) that "Thus, the improved SMB simulation in OPT-12L is achieved through compensation of errors".

12. In Table 3, consider also showing areal-mean biases.

13. line 705: "Metamorphism, dust . . . are ignored". In fact, snow metamorphism and dust are included in the model, albeit in a crude manner, through the snow aging parameterization.

14. I strongly appreciate the existence of Appendix A. However, to make it easier to use, I recommend ordering the list alphabetically. Say, first the quantities with Greek letters, followed by the quantities in the ordinary (i.e., Latin) letters in alphabetic order.

**Technical and language corrections**

1. line 310: should "required to" be "available to", "used to", "consumed in" etc.?

2. line 341: The latter "thickness" should be "thicknesses".

3. line 343: something missing before $\left(\delta_i, \frac{z_{snow}}{12}\right)$. "Min"?

4: Equations (24), (31) and (32): Consider changing the notation so that the melt terms (currently $S_{melt}$ and $I_{melt}$ ) are replaced with e.g. $M_{snow}$ and $M_{ice}$, to be consistent with the other terms in Eq. (32), which have phase indicated in the subindex.

5. As a follow-up comment on Eq. (32), according to Appendix A, the terms on the right hand side have different units: either "m" (for $P_{snow}$ and $P_{rain}$), "kg m$^{-2}$s$^{-1}$" (for $S_{snow}$), and "kg m$^{-2}$" for $S_{melt}$ and $I_{melt}$. Everything cannot be correct.

6. lines 411: "De Ridder and Schayes, 1997" is missing from the reference list.

7. line 516: Add "(e)" at the end of the figure caption.

8. Some of the titles on top of Figs. 2–6 have the word "differences" and some don't. Please harmonize them.

9. Appendix A: For $\delta_c$ and $P_{snow}$, units are given as "m". This is ambiguous. Are they water-equivalent values or not? If yes, "kg m$^{-2}$" would be better. For consistency, also for $P_{rain}$.

---

## Author Response (AR1)

We would like to thank the anonymous reviewer 1 for taking the time to proofread and provide insightful comments on the manuscript. We have done our best to address all comments in the revised version so as to improve the overall quality of the manuscript in line with the reviewer's recommendations. Our responses are reported below in blue.

**General comments**

This paper consists in essence of two parts. First, it provides a model description, documenting the Explicit Snow (ES) scheme implemented in the version of the ORCHIDEE land surface model used in the IPCL-CM Earth System Model for IPCC AR6, and the changes made to apply the ES scheme to glaciers and ice sheets, also including an increase in its vertical resolution. Second, the enhanced version of the model, named ORCHIDEE-ICE, is applied to the computation of snow mass balance (SMB) and its components over the Greenland ice sheet, in experiments forced with input data from the regional MAR model. With default parameter settings, the model clearly overestimates the SMB compared to MAR, and the improvement of vertical resolution reduces this bias only slightly. Good agreement with the MAR results for the SMB components is achieved by tuning the snow albedo parameterization, but at the cost of underestimated surface albedo over Greenland.

Overall, this paper paves the way towards including ice-sheet surfaces in the land surface module of the IPSL-CM ESM. At the same time, it provides a cautionary story, as satisfactory simulation of SMB is only achieved when the snow surface albedo is biased low.

I think this paper is a useful contribution to the topic, and it is generally well written. Therefore, I recommend publication of the paper in The Cryosphere subject to the relatively minor corrections/clarifications listed below.

**Specific comments**

1. lines 78–80: I think it would be worth mentioning, as an example of a relatively sophisticated GCM snow scheme, the Community Land Model (Lawrence et al. 2019) which includes the SNICAR scheme (Flanner and Zender 2005; 2006) and is employed at least in CESM and NorESM. This modelling system simulates prognostically snow density, grain size, liquid water and absorbing aerosols in a multi-layer snowpack, and computes snow albedo and absorption of solar radiation within the snowpack based on snow grain size and aerosol concentrations. Recently, SNICAR has been extented to also compute the albedo of glacier ice (Whicker et al. 2022), although (to my knowledge) it has not yet been coupled with a glacier model.

Thank you very much for drawing our attention to the SNICAR model implemented in CESM. We mentioned the modelling frameworks of Lawrence et al. (2019) and Flanner and Zender (2005,2006). We propose to change the sentence (lines 80-81 in the original version of the manuscript): *However, due to their high computational cost, they are not often used in ESMs, despite a few rare exceptions such as the work of Punge et al. (2012) based on the implementation of a detailed snow model (Brun et al., 1992) in the atmospheric model LMDZ4 (Hourdin et al., 2006), or the Community Land Model (CLM) which includes the snow radiative transfer scheme SNICAR (Flanner and Zender, 2006) and a snow model simulating a variety of key snow processes such as the metamorphism (Lawrence et al., 2019, He et al., 2024).*

2. lines 162–163. I'm puzzled about the sign convention here. Firstly, what is actually meant by saying that "$G_{surf}$" is computed negatively"? That $G_{surf}$ is negative when the net energy flux is directed downwards, and positive when it is directed upwards? Second, judging by Eqs. (3)

and (4), Hs and H$_L$ are positive upwards. So if Hs and H$_L$ increase, more energy is directed away from the surface which means that according to Eq. (1), $G_{surf}$ becomes more negative, which implies (according to line 163) that there is more warming of the surface?! This does not sound physically correct. Please, check and explain carefully the sign convention used in Eqs. (1–4). A graph showing which terms are positive upwards or downwards could be helpful.

*$G_{surf}$ is positive when the net energy flux is directed downwards and negative when it is directed upwards. On the other hand, Hs and H$_L$ are positive when they warm the atmosphere. As a result, when the turbulent heat fluxes are negative (i.e., warm the surface) while their absolute values increase, $G_{surf}$ also increases. This is what is written in Eq. (1). We guess that the confusion comes from the sentence "$G_{surf}$ is computed negatively when it cools the atmosphere". We acknowledge that it is a bit puzzling. We hope that we made the text clearer by replacing this sentence with "$G_{surf}$ is positive when it warms the soil".*

3. 3. Line 210: "$\delta c$ is the critical value of solid precipitation necessary for resetting the snow age to zero". This is only roughly true. For very cold temperatures, setting P = $\delta c$ results in $\tau_{snow}$ reduced by a factor of e, while for warm temperatures it may actually become negative.

*Although strictly a simplification, we believe this covers, in essence, what occurs in the model. Indeed, snow age is reduced by a factor 1/e at each time step. However, as the ORCHIDEE time step is 30 mn, snow age is almost zero in only a few time steps. For example, if $\tau_{snow}$ is set to 40 days before the snow fall event, it will be equal to $10^{-3}$ days in only 6 time steps (3 hours) as soon as $P_{snow} = \delta_c$. We changed the sentence you refer to in: As the ORCHIDEE time step is fixed to 30 mn,, the snow age is almost zero in a few time steps.*

*Moreover, we have to stress that $\tau_{snow}$ cannot become negative as surface temperature cannot be greater than 0°C over snow covered areas. Consequently, $g_{temp}(T_{surf})$ defined by Eq. (10) is always positive or zero, and so is snow age.*

4. Section 2.1.2: What is assumed about the vertical distribution of solar radiation absorbed by snow? And in ice (Section 2.2.2)? I'm getting the impression that all energy is deposited on the surface, and it is then distributed in snow only through diffusion, but it would be helpful to state this explicitly.

*We agree with this remark. Indeed, solar absorption is not accounted for in the ES model. We are aware that this is a crude simplification that may affect the accuracy of the model. However, as long as a more physically-based albedo scheme is not implemented in the model, we do think that this approximation is justified. In fact, solar absorption is highly dependent on snow optical properties which are themselves dependent on snow grain size. Since metamorphism is not explicitly represented in the model, we chose to ignore the absorption of solar energy in the snowpack. However, as specified in our responses to Reviewer 2, we are currently carrying out developments to implement a new snow spectral albedo model accounting for aerosols (light absorbing particles). However, this new model (Krishnakumar et al., 2024) is still under evaluation and is not available for this work. Its implementation in our ORCHIDEE-ICE model is the subject of very near future work.*

*Ignoring light absorption has impacts on albedo and therefore on the melting of snow and ice. This may be one of the main reasons why we have been compelled to reduce albedo. We followed the recommendations by stating explicitly in Section 2.1.2 that solar absorption in deep snow layers is not accounted for and by adding a comment on the related potential effect on the albedo: Solar absorption is not accounted for in the snow model. All incoming solar energy is therefore deposited at the snow surface and distributed in deeper layers through heat conduction.*

5. line 303: (Eq. 21): Does this indeed mean that the temperature of a snow layer only depends on the temperature of the layer above, and not on the temperature of the layer below? I would expect heat diffusion to work in both directions. The same question goes for ice (Eq. 27).

In equation (27), the temperature of layer $i+1$ is a function of the temperature of layer $i$ and the two coefficients $C_{gr\_snow}$ and $D_{gr\_snow}$. These coefficients were themselves calculated using the same numerical scheme as that used to calculate the temperature in the soil and published by Wang et al. (2016, see Appendix A herein). This simplified writing actually hides a complex calculation (through the determination of the coefficients $C_{gr\_snow}$ and $D_{gr\_snow}$) in which the temperature at the interface between two layers is calculated as a linear interpolation method according to the two nearest nodes (middle of the layers). Diffusion therefore takes place in both directions. This has been be clarified in the revised version: *The numerical scheme is similar to the one presented in Wang et al. (2016, see Appendix A therein) in which the temperature at the interface between two layers is calculated as a linear interpolation according to the two nearest nodes (middle of the layers). Diffusion therefore takes place downward and upward.*

6. Lines 394–395: "we also enhanced snow ageing by a factor of two in case of a rainfall event". This sounds rather ad-hoc, and furthermore, it is not clear what it actually means. I think you should report, with an equation, how rainfall impacts snow age ($\tau_{snow}$) in the model.

You are right, this was not very clear. In fact, in case of rainfall, we simply increase the function $f_{age}$ by a factor 2 ($f_{age} = f_{age}$ x 2). We agree that this is a completely ad-hoc parameterization. It has been introduced in the model to account for the effect of rain-on-snow events. Such events accelerate metamorphism and densification (Marshall et al., 1999), thereby lowering the albedo (Yang et al., 2023). To support this statement, we added the following sentences: *In case of rain-on-snow events, we also enhanced snow ageing by a factor of two ($f_{age} = f_{age}$ x 2). Although it sounds somewhat arbitrary, we introduced this parameterization in the model to account for the effect of such events on metamorphism and densification (Marshall et al., 1999), thereby lowering the albedo (Yang et al., 2023).*

7. lines 444–445: "Reducing the root-mean-square error (RMSE) by ~ 22%." Which quantity are you referring to? The RMSE in albedo? Also, Raoult et al. (2023) reported this number to be "over 25%".

Indeed, the RMSE refers to the albedo, and there was a typo error. In the revised manuscript we replaced the sentence you are referring to by: *"In doing this, they also succeeded in improving the model-data fit over the whole between the ORCHIDEE albedo and MODIS retrievals by reducing the root-mean-square error (RMSE) by ~25%".*

8. line 455: Which parameter(s) were the target in the optimization? Surface albedo or something else (SMB, runoff?). This seems to be said on line 605, but it should be reported already here.

The targets of the optimization are the SMB and its components (mainly runoff). This is achieved through the adjustment of albedo parameters in the range reported in Table 1 so as to lower the albedo, increase the runoff and decrease the SMB compared to the STD-3L experiment. To clarify this, we changed the sentence to which you refer to specify that the model performance is evaluated in terms of SMB and its components:

*"Therefore, using the new ORCHIDEE-ICE model version, we adopted a manual tuning approach (i.e., trial and error method) to adjust the albedo parameters (OPT-12L experiment). This procedure consists of 1/ changing the parameter values, the new values being taken from the range reported in Table 1, 2/ running the model with the new parameter values, 3/*

*evaluating the model performance (**in terms of SMB and its components**) using statistical criteria (e.g., RMSE **between MAR and ORCHIDEE-ICE**) and 4/ repeating steps 1/ to 3/ until an acceptable calibration is obtained".*

9. lines 534–535: This should be the other way round: "~ 11% lower and ~ 35% higher".

Yes, indeed. This has been corrected in the revised version.

10. lines 585–586. This sentence is not very clear. Does the implementation of ice layers make runoff smaller or larger?

We mean with lines 585-586 that accounting for ice layers makes the runoff larger. The sentence has been removed to rather insist on the impact of albedo and surface temperature. Indeed, in the southern part of GrIS, the albedo computed in ASIM-12L is slightly higher compared to MAR and even MODIS, with the consequence of limiting the temperature and thus the runoff. This has been mentioned in the revised version: *In addition, the simulated SMB (466 Gt yr$^{-1}$) has even been slightly degraded (Figs. 5a and 5d) due to negative temperature anomalies in central Greenland extending until the southern tip (Fig. 7c) resulting from slightly higher albedo values compared to MAR and MODIS (Figs 6a, 6i).*

11. lines 690: I would suggest to conclude this paragraph by saying explicitly (e.g.) that "Thus, the improved SMB simulation in OPT-12L is achieved through compensation of errors".

We mentioned the compensation of errors in the revised version. However, we do think that albedo reduction can compensate for missing processes such as the lack of an explicit representation of metamorphism which decreases the albedo, the non-inclusion of the penetration of incident solar energy into the snowpack and the fact snow-atmosphere feedbacks are ignored. We therefore added the following sentences: *Despite these encouraging results, it is important to underline that the improved SMB simulation in OPT-12L is achieved through the albedo reduction, and therefore, to some extent, come from error compensation. However, the reduced albedo also makes it possible to compensate for the effect of some missing mechanisms, such as the lack of consideration of snow-atmosphere interactions or the absence of an explicit representation of snow metamorphism, which has a direct impact on the density profile, the albedo itself and the temperature profile.*

12. In Table 3, consider also showing areal-mean biases.

We included in the revised manuscript the new Table 3 with areal mean biases (see below):

**Table 3 (revised):** Albedo RMSE values (column 1) and areal mean bias (column2) for the MAR, STD-3L, STD-12L, ASIM-12L and OPT-12L experiments compared to MODIS. Columns 3 and 4: same as columns 1 and 2 respectively for the ORCHIDEE-ICE experiments compared to MAR. All values are averaged over the 2000-2017 period.

| Experiments | RMSE (w.r.t MODIS) | Areal mean bias (w.r.t MODIS) | RMSE (w.r.t MAR) | Areal mean bias (w.r.t MAR) |
|---|---|---|---|---|
| MAR | 0.076 | - 0.005 | | |
| STD-3L | 0.098 | - 0.047 | 0.055 | - 0.042 |
| STD-12L | 0.097 | - 0.051 | 0.058 | -0.047 |
| OPTinit-12L | 0.072 | 0.001 | 0.051 | 0.006 |

| | | | |
|---|---|---|---|
| OPT-12L | 0.111 | - 0.008 | 0.092 | - 0.047 |

13. line 705: "Metamorphism, dust . . . are ignored". In fact, snow metamorphism and dust are included in the model, albeit in a crude manner, through the snow aging parameterization.

Yes, we agree. However, it would be more correct to say the effect of metamorphism on the albedo is accounted for through snow ageing parameterization. We added in the Discussion section: *snow metamorphism is only accounted for through snow ageing*

14. I strongly appreciate the existence of Appendix A. However, to make it easier to use, I recommend ordering the list alphabetically. Say, first the quantities with Greek letters, followed by the quantities in the ordinary (i.e., Latin) letters in alphabetic order.

Table 1 has been rearranged to make it easier to use. It now follows the alphabetical order as you suggest.

**Technical and language corrections**

1. line 310: should "required to" be "available to", "used to", "consumed in" etc.?

This has been changed in "available to"

2. line 341: The latter "thickness" should be "thicknesses".

Corrected

3. line 343: something missing before $\left(\delta_i, \frac{z_{snow}}{12}\right)$ "Min"?

Yes, you are right. "Min" are now added to the equation

4: Equations (24), (31) and (32): Consider changing the notation so that the melt terms (currently $S_{melt}$ and $I_{melt}$) are replaced with e.g. $M_{snow}$ and $M_{ice}$, to be consistent with the other terms in Eq. (32), which have phase indicated in the subindex.

The terms are now replaced in the equations and in Table 1 accordingly

5. As a follow-up comment on Eq. (32), according to Appendix A, the terms on the right hand side have different units: either "m" (for $P_{snow}$ and $P_{rain}$), "kg m$^{-2}$s$^{-1}$" (for $S_{snow}$), and "kg m$^{-2}$" for $S_{melt}$ and $I_{melt}$. Everything cannot be correct.

Yes, this is a good remark. We reported the units in Table 1 as they appear in the netcdf output files. Appropriate unit conversion has been done for the model results analysis. However, for more consistency, we homogenized them in the main text, the equations and in Table 1 (in kg m$^{-2}$ s$^{-1}$).

6. lines 411: "De Ridder and Schayes, 1997" is missing from the reference list.

The reference has been added to the reference list (see the complete references) at the end of our responses.

7. line 516: Add "(e)" at the end of the figure caption.

This was an omission. Thank you for this remark.

8. Some of the titles on top of Figs. 2–6 have the word "differences" and some don't. Please harmonize them.

Figures 2 to 4 (original manuscript) represent the raw distributions (absolute values) of runoff, sublimation and refreezing. Conversely, Figures 5b to 5e display the SMB differences between

ORCHIDEE and MAR, while Figure 5a is for the raw SMB distribution of MAR. We acknowledge that the titles of the figures may be confusing although everything is explained in the figure caption. To avoid any further confusion, we have split the figure into two parts: Raw distribution for the MAR SMB on the left-hand side and SMB differences on the right-hand side. The problem also arises for Figures 6 and 8 as Fig. 6a and Fig. 8a display the MAR and MODIS albedo respectively while the other panels represent the differences between the simulated albedo and the MAR albedo (Figs 6b-6e) or between the simulated albedo (MAR and ORCHIDEE-ICE) and the MODIS albedo (Figs 8b-8f). In the revised version, we joined Figures 6 and 8 in a single figure (which is now Figure 6) with the raw distributions of the albedo from MAR and MODIS displayed on the left, and the albedo differences between ORCHIDEE-ICE and MAR and between ORCHIDEE-ICE and MODIS on the right. For each ORCHIDEE-ICE experiment, we also added in the Supplement, the raw distributions of the albedo (Fig. S5), the SMB (Fig. S4), and the distributions of the differences with MAR of runoff (Fig. S1), refreezing (Fig. S2) and sublimation (Fig. S3).

9. Appendix A: For $\delta_c$ and $P_{snow}$, units are given as "m". This is ambiguous. Are they water-equivalent values or not? If yes, "kg m$^{-2}$" would be better. For consistency, also for $P_{rain}$.

Yes, these are water equivalent values per unit of time. All the units are now homogenized in Table 1 and in the main text (see also our response to your comment numbered 5).

**References:**

De Ridder, K, and Schayes G: The IAGL land surface model: Journal of Applied Meteorology, 36, 167-182, doi:10.1086/451461, 1997.

Krishnakumar, S., Albani, S., Ménégoz, M., Ottlé, C., & Balkanski, Y.: Influence of aerosol deposition on snowpack evolution in simulations with the ORCHIDEE land surface model (No. EGU24-8749). Copernicus Meetings, 2024.

Marshall, H.P., Conway, H., Rasmussen, L.A.: Snow densification during rain, Cold Regions Science and Technology, 30, 35-41, doi: 10.1016/S0165-232X(99)00011-7, 1999.

Wang, F., Cheruy, F., and Dufresne, J.-L.: The improvement of soil thermodynamics and its effects on land surface meteorology in the IPSL climate model, Geosci. Model Dev., 9, 363–381, https://doi.org/10.5194/gmd-9-363-2016, 2016.

Yang, Z., Chen, R., Liu, Y., Zhao, Y., Liu, Z., & Liu, J. (2023). The impact of rain-on-snow events on the snowmelt process: A field study, Hydrological Processes, 37(11), e15019; doi: 10.1002/hyp.15019, 2023.

**Responses to reviewer 2**

We would like to thank the anonymous reviewer 2 for taking the time to proofread and provide insightful comments on the manuscript. We have done do our best to address all comments in the revised version so as to improve the overall quality of the manuscript in line with the reviewer's recommendations. Our responses are reported below in blue.

This paper describes new modeling of Greenland snow and ice in the ORCHIDEE surface model. It is well written and pleasant to read. The work is substantial and certainly deserves to be published. The idea of using MAR as a forcing and comparing it to its outputs is very pertinent. Nevertheless, it seems to me that certain modeling choices need to be discussed and model validation a little improved.

**Major comments:**

1 - For Figures 1 to 4, you give the raw distributions of MAR and each simulation. But to show differences would be also interesting. In addition, for the others (Figure 5 to 8), you give the differences, but not the raw distributions. As it is possible for you to have additional information, it would be nice to provide all the details, i.e. both the raw distributions and the differences for each variable analyzed (and thus for each figure 1 to 8).

We guess that the reviewer means Figures 2 to 5 instead of Figures 1 to 4, as Figure 1 is the flowchart showing how the new snow scheme works. Figures 2 to 4 display the raw distribution of runoff, sublimation and refreezing for MAR and the four ORCHIDEE simulations. Figure 5 represents the raw SMB distribution of MAR (panel a) and the SMB differences between the ORCHIDEE runs and MAR outputs (panels b, c, d and e). The choice of plotting the SMB differences was motivated by the fact that SMB differences with MAR were not clearly visible with the raw distributions. We recognize that this representation is a little confusing, especially as the title of the figure only refers to panels (b, c, d and e) and the previous Figures 2 to 4 show the raw spatial distributions of runoff, sublimation and refreezing. Therefore, in the revised manuscript, Figure 5 (MAR SMB) has been rearranged with the raw SMB MAR distribution on the left-hand side and the SMB differences on the right-hand-side. The SMB raw distributions of each ORCHIDEE-ICE experiment are now provided in the Supplement (Fig. S4). The Supplement also alludes the spatial distributions of the differences (w.r.t. MAR) in runoff, refreezing and sublimation (Figs S1 to S3). Please, note that additional figures in the Supplement are provided at the end of this reply.

Figures 6 and 8: The albedo differences are only for panels b, c, d, e (Fig. 6) and b, c, d, e, f (Fig.8). Figs 6a and 8a represent the summer albedo coming from the MAR simulation and MODIS retrievals respectively. In the revised version, we joined Figures 6 and 8 in a single figure (which is now Figure 6) with the raw distributions of the albedo from MAR and MODIS displayed on the left, and the albedo differences between ORCHIDEE-ICE and MAR and between ORCHIDEE-ICE and MODIS on the right. For each ORCHIDEE-ICE experiment, we also added in the Supplement, the raw distributions of the albedo (Fig. S5), the SMB (Fig. S4), and the distributions of the differences with MAR of runoff (Fig. S1), refreezing (Fig. S2) and sublimation (Fig. S3).

2 - By the way, regarding this validation, why not provide the PDF of each quantity, i.e. compare the MAR (or MODIS) PDF with the ORCHIDEE PDFs ? When you look at Figures 1 to 8, it's hard to see whether one version is much better than another. An objective method of comparison is missing. Comparing PDFs could be a solution but there is perhaps another ways.

Following your suggestion, we plotted the PDFs for SMB, runoff, sublimation, refreezing and albedo to see what additional information they could bring to the analysis.

As ORCHIDEE has been forced by MAR outputs, the simulated ORCHIDEE SMB is very close to the SMB MAR in the GrIS interior. In other words, SMB differences mainly occur at the periphery of the ice sheet due in great part to differences in runoff and to a lesser extent in refreezing. As a result, plotting the PDFs for SMB, runoff and refreezing does not lead to very clear conclusions, since the MAR and ORCHIDEE PDFs are very close to each other. To better highlight the differences between both models, we have chosen to keep only the grid points for which the altitude is less than 1600 m. This altitude threshold corresponds to the maximum extent of areas experiencing ablation in the OPT-12L experiment. It is important to note that model outputs for sublimation have not been filtered at 1600 m, as sublimation/condensation occurs over the entire Greenland ice sheet, hence all grid points have been considered.

These distributions show that negative SMB values in the STD-3L experiment are less frequent than in the MAR signal and, conversely, that positive values are more common (Figs. S6 and S7, in the Supplementary Section). Moreover, positive SMB values are greater in the STD-3L experiment than in MAR. The same remarks can be made about STD-12L and ASIM-12, despite the differences with MAR being less pronounced. By contrast, the MAR and OPT-12L distributions are almost similar as shown by the Cramer Von Mises (CVM, Anderson, 1962; see also Table S1) statistical test and the related p-value (= 0.11).

Regarding runoff, sublimation and refreezing, the peaks of the distributions in the STD-3L, STD-12L and ASIM-12L experiments are shifted towards values lower than in the MAR distribution. These results were expected as the amount of runoff and, therefore, of refreezing are smaller in ORCHIDEE-ICE. In the same way, negative values found in the sublimation distributions correspond to a larger amount of condensation. Conversely, the SMB, the runoff and sublimation distributions of the OPT-12L experiments have significant similarities with the corresponding MAR distributions.

However, only the SMB OPT-12L distribution has a p-value greater than 0.05. This can be explained by the fact that ORCHIDEE -ICE was forced by MAR outputs and, as such, part of the SMB signal (i.e., accumulation) comes from MAR even in the ablation areas whose elevation is most often less than 1600 m. Nevertheless, values of the CVM tests clearly show that scores are improved between the STD-3L and the OPT-12L experiments (see Table S1), despite the fact that the obtained p-values (almost zero) cannot allow to conclude about the similarities between MAR and ORCHIDEE-ICE distributions (except for the OPT-12L SMB). These results confirm our previous conclusions presented in the main text and deduced from Figures 2 to 5 and Table 2.

We agree with the fact that the original version of the manuscript lacked statistical tests. However, as these tests do not provide new insights, we plan to include the above comment, the PDF figures and the CVM tests (Table S1) in the Supplementary Section.

We also provide the PDF for the albedo (see Fig. S8 and Table S2), although the ORCHIDEE-ICE distributions exhibit large differences with both the MAR and MODIS distributions, the former being shifted towards lower albedo values. An exception concerns ASIM-12L and MODIS. This result was expected as the albedo parameters used in ASIM-12L have been calibrated against MODIS retrievals.

Please note that the Supplementary Material can be found at the end of this reply.

Table S1: CVM test values for the SMB, runoff, sublimation and refreezing PDFs of the ORCHIDEE-ICE experiments compared to the corresponding MAR distribution. The corresponding p-values are reported in brackets.

| Experiments | SMB | Runoff | Sublimation | Refreezing |
|---|---|---|---|---|
| STD-3L | CVM = 25.09 [$3.20\ 10^{-9}$] | CVM = 34.25 [$6.36\ 10^{-9}$] | CVM = 170.62 [$5.55\ 10^{-8}$] | CVM:53.85 $6.70\ 10^{-9}$ |
| STD-12L | CVM = 12.73 [$3.07\ 10^{-9}$] | CVM = 15.81 [$4.66\ 10^{-9}$] | CVM = 162.01 [$6.36\ 10^{-9}$] | CVM = 22.82 [$7.64\ 10^{-9}$] |
| ASIM-12L | CVM = 6.16 $1.37\ 10^{-11}$ | CVM = 8.08 $1.37\ 10^{-11}$ | CVM = 368.87 $1.23\ 10^{-7}$ | CVM = 35.59 $1.15\ 10^{-8}$ |
| OPT-12L | **CVM = 0.33** **[0.11]** | CVM = 4.60 [$3.04\ 10^{-11}$] | CVM = 41.41 [$1.78\ 10^{-8}$] | CVM = 13.01 [$4.40\ 10^{-9}$] |

Table S2: CVM test values and the related p-values for the ORCHIDEE-ICE albedo compared to MAR and MODIS. Last line: CVM and p-values for the MAR albedo distribution compared to the MODIS distribution.

| Experiments | Albedo vs MAR | Albedo vs MODIS |
|---|---|---|
| STD 3L | CVM = 363.18 [$1.00\ 10^{-7}$] | CVM = 327.89 [$8.26\ 10^{-8}$] |
| STD-12L | CVM = 362.59 [$9.82\ 10^{-8}$] | CVM = 326.92 [$7.95\ 10^{-8}$] |
| ASIM-12L | CVM = 122.77 [$5.66\ 10^{-8}$] | CVM = 4.16 [$2.06\ 10^{-10}$] |
| OPT-12L | CVM = 320.44 [$1.10\ 10^{-7}$] | CVM = 296.28 [$7.24\ 10^{-8}$] |
| MAR | | CVM = 138.06 [$4.22\ 10^{-8}$] |

3 - It seems that spatial statistics (correlation, etc.) are missing for each field analyzed. In other words, two or three objective criteria to determine whether in Figure 2 (for example) the OPT-12L spatial distribution (e) is better than the others compared to MAR (a). That is, each panel (b, c, d, e) should have its spatial correlation (spatial rmse, etc.) with MAR. The fact that, for example, the spatial distribution of OPT-3L refreezing (Figure 3e) is closer to MAR (Figure 3a) is not trivial to see with the naked eye. Anyway, I hope this comment is understandable.

You lack objective statistical criteria in your assessment of all the figures showing comparisons of spatial distribution. The simple Table 2 obtained via a spatial average is not enough.

We agree with the reviewer on the fact that the comparison of the raw distributions is not always trivial. This is why we added the scatter plots displayed below (see Fig. 1 below, Fig. 8 in the revised version). The different points represent the daily values over the period 2000-2019 for each of the grid points. We believe that these new plots provide new insights and make the comments of our results more robust. In addition, we computed the areal mean biases (Tables 2 and 3 below) and the spatial RMSE (Table 2). Note that new information is provided in the new Table 2. Finally, we also added a new section (Section 4.3 in the revised manuscript) describing how we computed the statistical metrics.

[Figure]

Figure 1: Representation of the simulated SMB (1st row), runoff (2nd row), sublimation (3rd row) and refreezing (4th row) simulated with ORCHIDEE-ICE as a function of the same MAR variables: STD-3L (1st column), STD-12L (2nd column), ASIM-12L (3rd column) and OPT-12L (4th column). The different points represent the daily values over the period 2000-2019 for each of the grid points. The regression line is displayed in red (R is the regression coefficient) and the line y = x is in black.

Table 2 (revised): Simulated values of SMB, runoff, sublimation and refreezing (in Gt yr$^{-1}$) integrated over the entire Greenland ice sheet and averaged over the 2000-2019 period (column 2) and corresponding values (in mm day$^{-1}$) of the root-mean-square error (RMSE, column 3), areal-mean bias (column 4) and spatial RMSE (column 5) with respect to MAR outputs.

| Experiments | SMB (Gt yr$^{-1}$) | RMSE (in mm day$^{-1}$) | Areal mean bias (in mm day$^{-1}$) | Spatial RMSE (in mm day$^{-1}$) |
|---|---|---|---|---|
| **MAR** | **286** | | | |
| STD-3L | 504 | 0.976 | 0.351 | 3.05 |
| STD-12L | 450 | 0.786 | 0.264 | 2.81 |
| ASIM-12L | 466 | 0.706 | 0.290 | 2.60 |
| **OPT-12L** | **301** | **0.464** | **0.024** | **2.53** |
| ERA5-12L | 352 | | | |

| Experiments | Runoff (Gt yr$^{-1}$) | RMSE (in mm day$^{-1}$) | Areal mean bias (in mm day-1) | Spatial RMSE (in mm day$^{-1}$) |
|---|---|---|---|---|
| **MAR** | **375** | | | |
| STD-3L | 152 | 1.107 | - 0.357 | 3.16. |
| STD-12L | 205 | 0.922 | - 0.272 | 2.90 |
| ASIM-12L | 217 | 0.829 | - 0.254 | 2.64 |
| **OPT-12L** | **336** | **0.592** | **-0.063** | **2.54** |
| ERA5-12L | 273 | | | |

| Experiments | Sublimation (Gt yr$^{-1}$) | RMSE (in mm day$^{-1}$) | Areal mean bias (in mm day$^{-1}$) | Spatial RMSE (in mm day$^{-1}$) |
|---|---|---|---|---|
| **MAR** | **82** | | | |
| STD-3L | 33 | 0.021 | - 0.081 | 0.20 |
| STD-12L | 33 | 0.026 | - 0.079 | 0.20 |
| ASIM-12L | 5 | 0.050 | - 0.124 | 0.23 |
| OPT-12L | **52** | **0.050** | **- 0.049** | **0.27** |
| ERA5-12L | 89 | | | |

| Experiments | Refreezing (Gt yr$^{-1}$) | RMSE (in mm day$^{-1}$) | Areal mean bias (in mm day-1) | Spatial RMSE (in mm day$^{-1}$) |
|---|---|---|---|---|
| **MAR** | **186** | | | |
| STD-3L | 72 | 0.336 | - 0.183 | 1.25 |
| STD-12L | 104 | 0.269 | -0.131 | 1.13 |
| ASIM-12L | 90 | 0.313 | - 0.155 | 1.18 |
| **OPT-12L** | **158** | **0.240** | **-0.046** | **1.32** |
| ERA5-12L | 141 | | | |

**Table 3 (revised):** Albedo RMSE values (column 1) and areal mean bias (column2) for the MAR, STD-3L, STD-12L, ASIM-12L and OPT-12L experiments compared to MODIS. Columns 3 and 4: same as columns 1 and 2 respectively for the ORCHIDEE-ICE experiments compared to MAR. All values are averaged over the 2000-2017 period.

| Experiments | RMSE (w.r.t MODIS) | Areal mean bias (w.r.t MODIS) | RMSE (w.r.t MAR) | Areal mean bias (w.r.t MAR) |
|---|---|---|---|---|
| MAR | 0.076 | - 0.005 | | |
| STD-3L | 0.098 | - 0.047 | 0.055 | - 0.042 |
| STD-12L | 0.097 | - 0.051 | 0.058 | -0.047 |
| OPTinit-12L | 0.072 | 0.001 | 0.051 | 0.006 |
| OPT-12L | 0.111 | - 0.008 | 0.092 | - 0.047 |

4 - The modeling choices made could have been discussed. In particular, the parametrization of snow albedo. Moreover, I don't understand why this new parametrization compared to Wang et al. (2013) is in section 2.1 (existing parm) and not rather in section 2.2 (new param). This new parametrization is a bit outdated today when there are more robust parametrizations in land surface models accounting for spectral albedo and solar absorption calculation as in CLM with SNICAR (Flanner and Zender 2006) or ISBA-ES (Decharme et al. 2016). What's more, this more robust representation already exists for ES (Decharme et al. 2016). Why not use it ? This choice is debatable in view of the importance of albedo on the SMB. Please discuss about that.

This parameterization of the snow albedo is used in the standard ORCHIDEE model since the paper of Wang et al. (2013), this is why it is in section 2.1. We made it clearer in the revised version. We chose to drop the snow albedo ES parameterization after an exercise of intercomparison on sites and at a global scale against satellite data, which shows a better agreement with the simpler model, probably because it takes indirectly the vegetation impacts into account through the dependence of aging coefficients to plant functional types. We agree that over ice sheets, this choice is debatable, but we are presently carrying out developments in parallel around a new snow spectral albedo model accounting for aerosols (light absorbing particles). However, this new model (Krishnakumar et al., 2024) is still under evaluation and is not available for this work. Its implementation in our ORCHIDEE-ICE model is the subject of very near future work.

**Other comments:**

1 - Independent MAR observations of the Greenland SMB based on GRACE data could have been used in section 5.3 (Schlegel et al. 2016, Wang et al. 2024).

The GRACE space mission provides observations of variations in Earth gravity and therefore on variations in mass changes. For the Greenland ice sheet, mass changes result from SMB variations and from dynamic ice discharges. Determining mass changes related to ice dynamics requires the use of a dynamic ice sheet model forced, for example, by the simulated SMB. In fact, this is the approach followed by Schlegel et al (2016) who used the SMB computed with the MAR and RACMO models and the ISSM ice-sheet model. This is also the approach we intend to follow in the future within the framework of the coupling between the IPSL climate model and the GRISLI ice sheet model developed in our lab.

However, to address your comment and compare our results to observations independent from MAR, we used the 353 SMB measurements from the PROMICE (Programme for the Monitoring of the Greenland ice sheet) database (Machguth et al., 2016) and we computed the mean bias, the regression coefficient between PROMICE observations and model results and the RMSE. Results are reported in the table below. We also plotted the modelled SMB for OPT-12L, ERA5-12L and MAR for the grid points located closest to the observation sites as a function of the PROMICE SMB measurements (See Fig2 below, Fig. 12 in the revised).

[Figure]

**Figure 2:** Simulated SMB in the OPT-12L experiment and in MAR as a function of the observed SMB from the PROMICE network. As the observed SMB values are not all available over the same time interval, the measurements are given in meter water equivalent (mWE). 353 observations were available over the 2000-2019 period. Each simulated SMB value corresponds to the grid points located closest to the observation sites. The red line is the regression line with R being the correlation coefficient and the dashed black line indicates the line y = x.

As expected, MAR performs better than ORCHIDEE-ICE when compared with PROMICE. However, our results for the OPT-12L experiment are of the same order of magnitude as the MAR ones. Moreover, these results clearly indicate a strong reduction of both the bias and the RMSE in OPT-12L compared to the three other ORCHIDEE-ICE experiments. On the other hand, the ERA5-12L experiment also produces a good agreement with the observations despite a lower correlation coefficient than for MAR and OPT-12L. It is clear that the SMB simulated in the experiments forced by MAR is partly driven by the climate simulated by MAR itself (for the accumulation component). However, the results obtained with ERA5-12L clearly show that the behaviour of our model is consistent whatever the climate forcing used.

These results are now more extensively discussed at the end of Section 5.4 (formerly Section 5.3) as requested.

**Table 4 (new):** Comparison of the simulated SMB in MAR, STD–3L, STD-12L, ASIM-12L and OPT-12L with the SMB observations from the PROMICE network.

| Experiments | Bias (mWE) | Correlation | RMSE (mWE) |
|---|---|---|---|
| MAR | 0.14 | 0.86 | 0.82 |
| STD-3L | 0.94 | 0.67 | 1.70 |
| STD-12L | 0.68 | 0.73 | 1.43 |
| ASIM-12L | 0.74 | 0.75 | 1.33 |
| OPT-12L | 0.30 | 0.77 | 1.13 |
| ERA5-12L | 0.17 | 0.65 | 1.07 |

2 - From what I understand, some parameterizations that are in section 2.1 are new compared to Wang et al. (2013), and should therefore be in section 2.2.: snow fraction, albedo

As explained in our response to your Major Comment 4, this parameterization of the albedo has been modified since the paper of Wang et al. (2013). This is the same situation for snow fraction (see Eq. 6 in the manuscript). Both parameterizations were already implemented in the ORCHIDEE version used for CMIP6 simulations. This is why they are reported in Section 2.1 and not in Section 2.2 which only describes the new developments we made to apply the snow scheme to the ice sheets. In Section 2.1 (lines 134-135) we also specify: *"In this section, we provide an extensive description of the snow model including the main differences with the original ISBA-ES version (Wang et al., 2013)".*

3 - On the implementation of the ice layer (section 2.2.2), why didn't you use ES directly to model this. On line 293, you say that snow density is limited to 750kg/m³. But if you had raised this limit to 900 or 950kg/m³, it seems to me that all the "snow" equations converge to "ice" equations, at least that's what comes out when we compare equation 26 to 20, 27 to 21, etc. In theory, if ES is done right, snow that has reached a certain density should be able to become ice. It would then be sufficient to initialize the height and density of the snowpack accordingly (e.g. the last 6 layers with an ice density and a total height of 100m for example). I don't know if it's possible but this could be discussed.

The value of 750 kg m$^{-3}$ was chosen in the original ES version (and it has not been changed in ORCHIDEE-ICE) because compaction becomes slower above densities between 550 and 800 kg m$^{-3}$ (Maeno and Ebinuma, 1983). A critical value of 730 kg m$^{-3}$ has even been advanced by Maeno (1978). The slower compaction is due to the progressive disappearance of air spaces between the snow particles. When the snowpack attains such high densities, the transformation of firn into ice slows down dramatically. For example, Schwander and Stauffer (1984) report that in Greenland, the process does not exceed 80 years but ranges between ~20 and ~ 600 years in Antarctica, depending on the stations. As a result, to reduce the spin-up duration, we chose to prescribe ice layers. These are considered as an infinite reservoir and they are only used to compute the amount of the potential ice ablation.

To address your comment, we specified in Section 2.1.2: *This value [750 kg m$^{-3}$] was chosen because compaction becomes slower above densities between 550 and 800 kg m-3 due to the progressive disappearance of air spaces between the snow particles (Maeno and Ebinuma, 1983). A critical value of 730 kg m-3 has even been advanced by Maeno (1978).*

It is also worth mentioning that at the edges of the ice sheet, snow can be fully removed through seasonal melting giving rise to bare ice. In turn, ice can melt if the energy is sufficient. In real life, ice loss is replaced by the upstream ice flow. However, in the absence of a dynamic ice sheet model, this ice flow is not accounted for.

4 - Line 629 - 632: I understand here that the improved runoff modeling in OPT-12L would not be due to bias compensation. Well, I'm really not sure. What I understand from looking at your results is that to improve runoff compared to MAR, you need to set an albedo lower than MAR (Figure 6e), which inevitably induces a surface temperature (and surely an internal temperature of the snowpack) that is too high (Figure 7e) compared to MAR. I have the impression that this is also what Figure (8f) reveals. So, to obtain the same runoff than MAR, ORCHIDEE-ICE have to simulate a lower albedo than MAR to capture more energy. If it is true, it is perhaps due to the non-representation of solar absorption by snow or a poor simulation of snowpack density.

We recognize that this sentence implies something that is not quite right. What we had in mind was that the RMSE decreases between STD-3L and OPT-12L. So, if there is error compensation, it decreases from one experiment to the next. According to our RMSE values for SMB, runoff, sublimation and refreezing, the OPT-12L experiment better compares to MAR than any other ORCHIDEE-ICE experiment. Other statistical tests you requested confirm this statement, at least for SMB and runoff. For refreezing, the OPT-12L spatial RMSE has the highest value although the areal-mean bias has the lowest one (see new Table 2). In the same way, the regression line coefficient of the sublimation (see previous scatter plots) reflects a larger mismatch with MAR. In these cases, we agree on the fact that there is actually "compensation of errors". These new statistical tests are now presented in the revised manuscript and the limitations are also mentioned in Section 5.2 and in the Discussion section. To avoid confusion, we also removed the lines 630-631 from the original version stating that "the improvement in the runoff computation over the whole of the GrIS does not result from compensation biases".

On the other hand, we acknowledge that our model has missing mechanisms such as the transmission of solar absorption (which may lead to higher snowpack temperatures), or biases in the snow density (see next response). We are fully aware of these limitations and we extended the discussion to mention these aspects. However, we would like to remind you that our primary objective in this paper was to present the state-of-the-art performance of our model and its ability to represent the SMB and its components under prescribed conditions (i.e., prescribed albedo parameters leading to lower albedo). As mentioned above, a more physically based albedo scheme is currently under development and will be implemented in ORCHIDEE-ICE in the near future.

In the Discussion Section, we added the following paragraph:

*Taking into account the transmission of solar radiation within the snowpack can lead to a warming of the internal layers, with higher temperatures near the surface and lower temperatures at depth due to the exponential decrease in heat transfer. This results in a temperature gradient that influences the metamorphism of snow grains and thus accelerates densification (Colbeck, 1983). We showed that the ORCHIDEE-ICE temperatures inside the snowpack were higher than those simulated by the MAR model. A likely hypothesis to explain this behaviour relies on the reduction in albedo, which leads to excessively high surface temperatures. Given this observation, it seems unlikely that accounting for solar absorption may improve our results. However, it is important to note that heat transfer can promote snow*

*melting, which in turn can percolate at depth and refreeze, affecting both the runoff and the vertical structure of the snowpack through changes in density (Colbeck, 1983). Quantifying all these processes requires, therefore, the proper representation of solar absorption, which is itself strongly dependent on snow optical properties (Warren, 1982) and, therefore, on snow grain size (Libois et al., 2013). Since metamorphism is not explicitly represented in the model, we think that ignoring solar absorption is justified. However, a more pysically based albedo scheme accounting for light-absorbing particles and snow grain size (Kokhannovsky and Zege, 2004) will be implemented in the ORCHIDEE-ICE model in the near future.*

5 - This last remark also underlines the fact that other important variables concerning the internal properties of the snowpack could be shown/analyzed, such as the temperature and density of the simulated snowpack compared with MAR, etc. This would enable a better understanding of the processes involved in the improvements related to one or another process claimed by the authors.

As you pointed out, ignoring solar absorption may have impact on the vertical temperature profile. In our simulations this seems to be only true for temperatures below ~ -10°C as shown in Figure 3 below (Fig. 9 in the revised manuscript for depths of 50 cm and 1 meter). Indeed, temperatures above this threshold present a relatively good agreement with the simulated MAR temperatures at 1 and 2 meter-depth. As a result, runoff should not be so much impacted. Moreover, a more surprising result is that the temperature profile does not seem to be too much affected by the choice of the albedo parameterization (i.e., values of the albedo parameters), thereby limiting the impact on runoff.

Warmer temperatures in a given snowpack may modify the snow microstructure and the water vapor transfer between snow particles, as well as snow density. This process is parameterized in the snow model through the second term of Eq. (17). However, it becomes rapidly negligible for density above 150 kg m$^{-3}$. It is therefore likely that the effect of this parameterization on the albedo is not very significant at depths above 1 m as snow density exceeds 300 kg m$^{-3}$ (see Figure 3). Nevertheless, in a future study, it should be interesting to investigate thoroughly the extent to which snow temperature is affected by the parameterization of water vapor transfer.

The differences in the simulated densities between MAR and ORCHIDEE are mainly related to high values simulated by MAR (Fig. 4 below and Fig. 10 in the revised manuscript for depths of 50 cm and 1 meter). For example, snow density at 2 m depth does not exceed 600 kg m$^{-3}$ in OPT-12L, while it reaches values up to 800 kg m$^{-3}$ in MAR. Similar remarks can be made for 5 meter-depth (not shown). This result can be explained by the choice of a lower maximum snow density value used in ORCHIDEE-ICE (750 kg m$^{-3}$) and by the fact that snow metamorphism is not explicitly represented in our snow model. This limitation is now pointed out in the revised manuscript.

Moreover, to address more thoroughly your comment, we added a new section (Section 5.3 entitled "Vertical temperatures and density profiles")

[Figure]

Figure 3: Representation of the ORCHIDEE-ICE simulated snow temperatures at one-meter (left) and two-meter depth (right) as a function of MAR. The different points represent the daily values over the period 2000-2019 for each of the grid points. The regression line is displayed in red (R is the regression coefficient) and the line y = x is in black.

[Figure]

Figure 4: Representation of the ORCHIDEE-ICE simulated snow densities at one-meter (left) and two-meter depth (right) as a function of MAR. The different points represent the daily values over the period 2000-2019 for each of the grid points. The regression line is displayed in red (R is the regression coefficient) and the line y = x is in black.

**References**

Anderson, T. W: On the distribution of the two-sample Cramer von Mises criterion, The Annals of Mathematical Statistics, 33, 1148–1159, 1962.

Krishnakumar, S., Albani, S., Ménégoz, M., Ottlé, C., & Balkanski, Y. (2024). *Influence of aerosol deposition on snowpack evolution in simulations with the ORCHIDEE land surface model* (No. EGU24-8749). Copernicus Meetings.

Machguth, H., Thomsen, H. H., Weidick, A., Abermann, J., Ahlström, A. P., Andersen, M. L., Andersen, S. B., Björk, A. A., Box, J. E., Braithwaite, R. J., Bøggild, C. E., Citterio, M., Clement, P., Colgan, W., Fausto, R. S., Gleie, K., Hasholt, B., Hynek, B., Knudsen, N. T., Larsen, S. H., Mernild, S., Oerlemans, J., Oerter, H., Olesen, O. B., Smeets, C. J. P. P., Steffen, K., Stober, M., Sugiyama, S., van As, D., van den Broeke, M. R., and van de Wal, R. S.: Greenland surface mass balance observations from the ice sheet ablation area and local glaciers, J. Glaciol., 62, 861–887, https://doi.org/10.1017/jog.2016.75, 2016.

Maeno, N., Narita, H., and Araoka, K: . Measurements of air permeability and elastic modulus of snow and firn drilled at Mizuho station east Antarctica, Memoirs of National Institute of Polar Research, 10, 62-76, 1978.

Maeno, N., and Ebinuma, T.: Pressure sintering of ice and its implication to the densification of snow at polar glaciers and ice sheets, Journal of Physical Chemistry, 87, 4103-4110, 1983.

Schwander, J., and Stauffer, B.: Age difference between polar ice and the air trapped in its bubbles. Nature 311, 45–47, https://doi-org.insu.bib.cnrs.fr/10.1038/311045a0, 1984.

Wang, T., C. Ottlé, A. Boone, P. Ciais, E. Brun, S. Morin, G. Krinner, S. Piao, and S. Peng,: Evaluation of an improved intermediate complexity snow scheme in the ORCHIDEE land surface model, J. Geophys. Res. Atmos., 118, 6064–6079, doi:10.1002/jgrd.50395, 2013.

**Supplementary Materials**

Runoff differences 2000-2019 (mm day$^{-1}$)

[Figure]

Figure S1: Spatial distribution of the differences in the simulated runoff (in mm day$^{-1}$) between MAR and and the ORCHIDEE-ICE experiments: STD-3L (a), STD-12L (b), ASIM-12L (c) and OPT-12L (d).

**Sublimation differences 2000-2019 (mm day$^{-1}$)**

[Figure]

Figure S2: Same as Figure S1 for sublimation.

**Refreezing differences 2000-2019 (mm day$^{-1}$)**

[Figure]

Figure S3: Same as Figure S1 for refreezing.

SMB 2000-2019 (mm day$^{-1}$)

[Figure]

Figure S4: Spatial distribution of the simulated SMB (in mm day$^{-1}$) averaged over the 2000-2019 period for the STD-13L (a), STD-12L (b), ASIM-12L (b) and OPT-12L experiments.

**Summer albedo 2000-2017**

[Figure]

Figure S5: Spatial distribution of the summer (June-July-August) averaged over the 2000-2017 period for MODIS (a), MAR (b), STD-3L (c), STD-12L (d), ASIM-12L (e) and OPT-12L (f).

[Figure]

Figure S6: Probability density function of the STD-3L and OPT-12L experiments compared to the MAR distributions for SMB, runoff, sublimation and refreezing.

[Figure]

Figure S7: Probability density function of the STD-12L and ASIM-12L experiments compared to the MAR distributions for SMB, runoff, sublimation and refreezing.

[Figure]

Figure S8: Probability density function for the simulated albedo of the ORCHIDEE-ICE experiments compared to MAR.

---

## Author Response (AR2)

**Response to the reviewer's comments**

The article has been much improved and deserves to be published. I have only a few minor comments to make.

We are grateful to the reviewer for taking the time to review our manuscript a second time and for appreciating the work we have done to respond as well as possible to his initial comments. Below are our responses to the new comments (in blue). Excerpts from the text are italicized.

1/ You say "A likely hypothesis to explain this behaviour relies on the reduction in albedo, which leads to excessively high surface temperatures. Given this observation, it seems unlikely that accounting for solar absorption may improve our results." Here you're drawing conclusions from something you don't model. I'd prefer that last sentence to be deleted ("Given this observation, it seems...").

As recommended, we removed the sentence.

2/ You say that "Since metamorphism is not explicitly represented in the model,we think that ignoring solar absorption is justified." This sentence should be more measured and should be addressed with more cautions. Some model represents solar absorption without representing snow metamorphism (Decharme et al 2016 is a good example of this that is often cited in your study). Perhaps only say : "Since metamorphism is not explicitly represented in the model, we made the assumption that representing solar absorption was not a priority for our modeling, even if this choice is debatable."

The sentence cited by the reviewer has been changed in: *"Since metamorphism is not explicitly represented in the model, we assumed that representing solar absorption was not a priority in our modeling approach, even if this choice is debatable".*

3/ And after you say "a more physically-based albedo scheme accounting for light-absorbing particles and snow grain size will be implemented in the ORCHIDEE-ICE model in the near future." So, if you use snow grain size for albedo, why not for deep solar absorption? In your discussion, you seem to separate the processes of solar absorption and albedo. But they're closely linked... and one doesn't go without the other in nature for materials like water, ice or snow...".

In light of the reviewer's comment, we realise that we were not clear enough in our explanations. To avoid any confusion, we modified the sentence: *"However, a more pysically based albedo scheme accounting for light-absorbing particles and snow grain size (Kokhannovsky and Zege, 2004) will be implemented in the ORCHIDEE-ICE model in the near future".*

The modified sentence now reads as: *"However, in the near future, a more sophisticated albedo scheme based on a transfer radiative model accounting for light-absorbing particles and snow grain size (Kokhannovsky and Zege, 2004) will be implemented in the ORCHIDEE-ICE model. This will allow to represent the backward and forward scattering processes as well as light absorption".*